

# Lightning-ignited wildfires and long-continuing-current lightning in the Mediterranean Basin: Preferential meteorological conditions

Francisco J. Pérez-Invernón[1], Heidi Huntrieser[1], Sergio Soler[2], Francisco J. Gordillo-Vázquez[2], Nicolau Pineda[3,5], Javier Navarro-González[4], Víctor Reglero[4], Joan Montanyà[5], Oscar van der Velde[5], and Nikos Koutsias[6]

[1]Deutsches Zentrum für Luft- und Raumfahrt, Institut für Physik der Atmosphäre, Oberpfaffenhofen, Germany
[2]Instituto de Astrofísica de Andalucía, CSIC, Glorieta de la Astronomía s/n, 18008 Granada, Spain
[3]Meteorological Service of Catalonia, Carrer Berlín 38–46, 08029 Barcelona, Spain
[4]Image Processing Laboratory, University of Valencia, Valencia, Spain
[5]Lightning Research Group, Technical University of Catalonia, Campus de Terrassa, Edifici TR1, Carrer Colom 1, Terrassa, 08222 Barcelona, Spain
[6]Department of Environmental Engineering, University of Patras, G. Seferi 2, Agrinio GR-30100, Greece

**Correspondence:** Francisco J. Pérez-Invernón (FranciscoJavier.Perez-Invernon@dlr.de)

**Abstract.** Lightning is the major cause of natural ignition of wildfires worldwide and produces the largest wildfires in some regions. Lightning strokes produce about 5% of forest fires in the Mediterranean basin and are one of the most important precursors of the largest forest fires during the summer. Lightning-ignited wildfires produce significant emissions of aerosols, black carbon and trace gases, such as $CO$, $SO_2$, $CH_4$ and $O_3$, affecting air quality. Characterization of the meteorological and

cloud conditions of lightning-ignited wildfires in the Mediterranean basin can serve to improve fire forecasting models and to upgrade the implementation of fire emissions in atmospheric models.

This study investigates the meteorological and cloud conditions of Lightning-Ignited Wildfires (LIW) and Long-Continuing-Current (LCC) lightning flashes in the Iberian Peninsula and Greece. LCC lightning and lightning in dry thunderstorms with low precipitation rate have been proposed to be the main precursors of the largest wildfires. We use lightning data provided by

the World Wide Lightning Location Network (WWLLN), the Earth Network Total Lightning Network (ENTLN) and the Lightning Imaging Sensor (LIS) onboard the International Space Station (ISS) together with four databases of wildfires produced in Spain, Portugal, Southern France and Greece, respectively, in order to produce a climatology of LIW and LCC lightning over the Mediterranean basin. In addition, we use meteorological data provided by the European Centre for Medium-Range Weather Forecasts (ECMWF) ERA5-reanalysis data set and by the Spanish State Meteorological Agency (AEMET) together with the

Cloud Top Height (CTH) product derived from Meteosat Second Generation (MSG) satellites measurements to investigate the meteorological conditions of LIW and LCC lightning. According to our results, LIW and a significant amount of LCC lightning flashes tend to occur in dry thunderstorms with weak updrafts. Our results suggest that lightning-ignited wildfires tend to occur in high-based clouds with a vertical content of moisture lower than the climatological value, as well as with a higher temperature and a lower precipitation rate. Meteorological conditions of LIW from the Iberian Peninsula and Greece

are in agreement, although some differences possibly caused by highly variable topography in Greece and a more humid en-





vironment are observed. These results show the possibility of using the typical meteorological and cloud conditions of LCC lightning flashes as proxy to parameterize the ignition of wildfires in atmospheric or forecasting models.

## 1  Introduction

Every second, about 44 lightning flashes occur around the globe (Christian et al., 2003; Cecil et al., 2014). Apart from being one of the major sources of nitrogen oxides ($NO_x = NO + NO_2$) in the troposphere [e. g., Schumann and Huntrieser (2007) and references therein], lightning discharges are the main precursors of natural forest fires worldwide (Komarek, 1964; Pyne et al., 1998; Latham and Williams, 2001). Lightning-Ignited Wildfires (LIW) produce significant emissions of aerosols, black carbon and some trace gases, such as CO, $SO_2$, $CH_4$ and $O_3$ [e. g., Schultz et al. (2008); Van der Werf et al. (2010); Huntrieser

et al. (2016); Van Der Werf et al. (2017)]. However, there are still noteworthy questions about the meteorological conditions that favor the occurrence of LIW, especially in Europe. For example, what is the relationship between the occurrence of dry thunderstorms and LIW fires in the Mediterranean basin? What is the role of Long-Continuing-Current (LCC) lightning flashes in the ignition?

The linkage between lightning and forest fires has been widely investigated by several authors in different regions using data
provided by aircraft campaigns, satellite observations and lightning-detection networks [e. g., Lyons et al. (1998); Anderson (2002); Stocks et al. (2002); Wotton and Martell (2005); Hall and Brown (2006); Fernandes et al. (2006); Kochtubajda et al. (2006); Lang and Rutledge (2006); Rosenfeld et al. (2007); Hall (2007); Altaratz et al. (2010); Dowdy and Mills (2012); Nauslar et al. (2013); Lang et al. (2014); Veraverbeke et al. (2017)]. In Canada, LIW account for 80% (Anderson, 2002) of total area burned, while in the western states of the US, the total area burned by LIW is about 60% (Nauslar et al., 2013).

However, European studies are rare. Lightning is the major cause of ignition in the European boreal forests (Granström, 2001; Larjavaara et al., 2005b, a; Granström and Niklasson, 2008; Rolstad et al., 2017) and one of the main cause of ignition in the Alps (Conedera et al., 2006; Moris et al., 2020). In the Mediterranean basin, lightning causes about 5% of the total number of forest fires (Vázquez and Moreno, 1998; Camia et al., 2010; Koutsias et al., 2013), while the rest of fires are man-caused. However, LIW are one of the most important precursors of the largest forest fires during the summer (Vázquez and Moreno,

1998; Badia et al., 2002; Amatulli et al., 2007; García-Ortega et al., 2011; Oliveira et al., 2012; San-Miguel-Ayanz et al., 2013; Ganteaume et al., 2013; San José et al., 2014). Contrary to fires with anthropological origin, LIW usually occur in remote areas and under extreme meteorological conditions that hinder the extinction work (Pineda et al., 2014; Pineda and Rigo, 2017). The important role of lightning in initiating the largest forest fires over the Mediterranean basin highlights the need to develop novel forecasting methods.

The ignition (fire triggering), survival (smoldering) and arrival (flaming combustion) of LIW (Anderson, 2002) is closely related with intense drought periods and high temperatures (Pineda and Rigo, 2017). The amount of precipitation at surface





(Colson, 1960; Hall, 2007) and the type of vegetation (Pineda et al., 2014; Gora et al., 2017; Baranovskiy and Yankovich, 2018) play a key role in the ignition of wildfires. Most of LIW occur in forest with conifer vegetation (Krawchuk et al., 2006; Reineking et al., 2010; Müller et al., 2013; Pineda and Rigo, 2017; Moris et al., 2020), where dry thunderstorms and strong winds favor the ignition, survival and arrival of fire (Rorig et al., 2007; Pineda and Rigo, 2017). The definition of dry thunderstorm is not homogeneous in the literature. Some authors define dry thunderstorms as storms with total precipitation below nearly 2.5 mm (e.g., Rorig et al., 2007). In this work, we define dry thunderstorms as thunderstorms with 1-hourly accumulated precipitation below the climatological value to investigate the conditions at the moment of ignition and right before possible survival. Some studies over the US suggest that dry lightning and LIW tend to occur in high-based clouds (clouds with high cloud base) (Nauslar et al., 2013). Furthermore, significant efforts have been done to investigate the meteorological conditions of fire-producing thunderstorms at ground and in the lower troposphere. However, European studies are rare. For example, Rorig et al. (2007) investigated the role of the instability and content of moisture at midlevel (850-500 hPa). Some research has been done so far on the meteorological conditions of fire-igniting thunderstorms near the upper troposphere, where electrification occurs. For example Wallmann (2004) and Nauslar et al. (2013), included the dynamics of the upper troposphere and the tropopause in the forecasting of dry thunderstorms over the United Stats.

Lightning discharges can be classified into Intra-Cloud (IC), Cloud-to-Cloud (CC) and Cloud-to-ground (CG) discharges. CG lightning discharges are initiated between two cloud regions with opposite sign electrical charges before escaping from clouds (Rakov and Uman, 2003). After the onset of the discharge, an advancing streamer corona guides the propagation of the leader, a high-temperature plasma channel. When the leader connects the ground to the main charged cloud layer, electrical charge travels across the channel, producing one or several Return Strokes (RS). The majority of the RS are composed by a highly impulsive electrical current with a duration less than 10 ms, producing electromagnetic radiation that can be detected thousands of kilometers away by Lightning Location Systems (LLS). However, some CG flashes have a continuing current phase lasting between tens to hundreds milliseconds. This type of CG lightning discharges are known as Long-Continuing-Current lightning flashes (LCC lightning flashes), being able to transport a significant amount of electrical charge between the cloud and the attachment point. The continuing current phase of the discharge emits radio signal in the Extremely Low Frequency (ELF) range of frequencies, preventing their detection by typical LLS, that are developed to report radio signals emitted by lightning discharges in the Very Low Frequency (VLF) range. However, the continuing phase of LCC lightning can be detected from optical measurements (Fuquay et al., 1967; Adachi et al., 2009). In this work, we use optical observations reported from the space-based instrument Lightning Imaging Sensor (LIS) to investigate the continuing current phase of lightning discharges over Europe (Bitzer, 2017). For a more extensive description of the physical characteristics of lightning discharges and their electromagnetic emissions, we refer to Rakov and Uman (2003),

Some evidence suggests that LCC lightning flashes are precursors of LIW. This was originally proposed by McEachron and Hagenguth (1942) working with laboratory sparks, who suggested that ignition by natural lightning is usually caused by a discharge having an unusual LCC phase. Later in 1967 this hypothesis was confirmed by Fuquay et al. (1967), who reported seven cases of fire-igniting lightning strokes with duration between 40 ms and 282 ms. Fuquay et al. (1967) and Adachi et al. (2009) showed that the optical signal emitted by lightning discharges can be related to the duration of the electrical discharge.





Bitzer (2017) reported the first tropical and mid-latitude climatology of LCC lightning discharges with a duration larger than 10 ms. Bitzer (2017) provided the total number of LCC lightning with duration up to 40 ms from optical lightning measurements reported by LIS onboard the Tropical Rainfall Measuring Mission (TRMM) satellite following a low-Earth orbit between 1997 and 2015, providing lightning measurements in the range of latitude between 35°N and 35°S. According to Bitzer (2017), LCC lightning discharges tend to occur in oceanic and winter thunderstorms, where the updrafts are weaker than in typical summer thunderstorms over land. Bitzer (2017) proposed that thunderstorms with weaker updrafts would produce small charging rates, allowing the charging process to develop larger charge regions before the onset of lightning and providing the discharge with more energy to be transferred. However, the TRMM satellite did not cover the European continent. LIS is now operating onboard the International Space Station (ISS), covering for the first time the Mediterranean basin and providing optical measurements of the duration of lightning pulses.

In this work, we investigate the meteorological characteristics of fire-producing thunderstorms and the electrical characteristics of the lightning-igniting fires in the Mediterranean basin. This approach will serve to improve fire forecasting methods and atmospheric models including fire-emissions. As we cannot directly connect LIW to LCC flashes using LLS, we search for shared meteorological conditions. We develop the first climatology of LCC lightning over Europe from the lightning data provided by LIS onboard the International Space Station (ISS-LIS) with a duration larger than 20 ms and its possible relationship with the occurrence of LIW. This duration is lower than the lowest duration of the continuing phase of a fire-igniting lightning reported by Fuquay et al. (1967) (40 ms), but enough to be considered as a LCC lightning flash. In particular, we focus our analysis on the Iberian Peninsula (Spain, Portugal and the Mediterranean France) and Greece. We do not include data over Italy because they were not accessible. We combine five fire databases from these countries with lightning measurements provided by the World Wide Lightning Location Network (WWLLN) and the Earth Network Total Lightning Network (ENTLN). We use meteorological data provided by the European Centre for Medium-Range Weather Forecasts (ECMWF) ERA5-reanalysis data set and by the Spanish State Meteorological Agency (AEMET). For the first time, we combine LCC lightning data provided by LIS onboard the International Space Station (ISS) with meteorological data sets.

## 2 Data and methodology

Table 1 shows the main characteristics of the data sets employed in this study. In this section, we describe each of the data set and the employed methodology to investigate the meteorological conditions of LIW and LCC lightning flashes.

### 2.1 Lightning measurements

We use ground-based lightning data provided by the Lightning Locations Systems WWLLN and ENTLN to search the lightning candidates for the forest fires in the periods 2009-2013 and 2014-2019, respectively. In addition, we use lightning optical measurements reported by ISS-LIS between 2017 and 2020 to investigate LCC lightning over Europe.

The ground-based WWLLN is composed by a global network of VLF sensors that can provide the position, time of occurrence and energy radiation by lightning discharges (Dowden et al., 2002; Rodger et al., 2005). WWLLN is more sensitive





**Table 1.** Datasets used in this study. References are given in the text of Section 2.

| Data set | Description | Region | Temporal coverage | Variables |
|---|---|---|---|---|
| WWLLN | VLF lightning data with a stroke DE raging between 8-13% | Mediterranean Basin | April 2009- December 2013 | Time, location, radiated energy of lightning strokes |
| ENTLN | VLF lightning data with a stroke DE about 57% | Mediterranean Basin | January 2014- December 2019 | Time, location, polarity and peak current of lightning flashes |
| ISS-LIS | Optical lightning data | Europe | March 2017 - September 2020 | Time and location of groups and lightning flashes |
| ERA5 | Reanalysis meteorological data | Mediterranean Basin | 2009-2020 | CAPE, CBH, horizontal wind components, hourly-accumulated precipitation, vertical profiles between ground and 200 hPa pressure level of the temperature, the relative humidity, the vertical velocity, the specific cloud ice water content, the specific cloud liquid water content, the specific rain water content and the specific snow water content. |
| PROBA-V and S3 OLCI time series | Classification of forest type | Mediterranean Basin | 2015 | Type of trees in forest |
| MSG CTH product | CTH from data acquired by SEVIRI | Mediterranean Basin | 2011-2020 | CTH |
| Echo top radar provided by AEMET | Echo top heights measurements (maximum height of the 12-dBZ echo) | Spain | 2015 | Echo top heights |
| Forest fires over Spain | Fire data provided by the Spanish Ministerio de Agricultura, Pesca y Alimentación | Spain | 2009-2015 (complete) and 2016-2017 (regional) | Location, time of detection and cause of ignition of 92187 fires (2009-2015). 3162 (2009-2015) are LIW (about 3.4%). |
| Forest fires over Portugal | Fire data provided by the Instituto da Conservaçao da Naturaleza e das florestas | Portugal | 2009-2015 | Location and time of detection of 163190 fires. Thereof 359 LIW filtered out (0.22%). |
| Forest fires over France | Fire data provided by the Prométhée database | Mediterranean France | 2009-2015 | Location, time of detection and cause of ignition of 797 fires. 36 are LIW (about 4.5%). |
| Forest fires over Greece | Fire data provided by the Hellenic Fire Brigade | Greece | 2017-2019 | Location and time of detection of 62690 fires. Thereof 1999 LIW filtered out (3.2%). |

to CG lightning discharges than IC, with a total global stroke Detection Efficiency (DE) between 2009 and 2013 of 8-13%

(Hutchins et al., 2012a; Rudlosky and Shea, 2013; Bitzer et al., 2016). The stroke location accuracy of WWLLN is between 5 km and 10 km, while the temporal accuracy is about tens of microseconds (Abreu et al., 2010; Rudlosky and Shea, 2013). In this work, we use the relationship between the energy of the strokes measured by WWLLN and the peak current of the discharge, as reported by Hutchins et al. (2012b) to estimate the peak current of the fire-igniting lightning.

The ground-based ENTLN is a global network composed by VLF sensors that provide the position, time of occurrence,

polarity and peak current of lightning strokes. ENTLN has a DE of about 90% for CG strokes over the US (Zhu et al., 2017; Lapierre et al., 2020) and a total global stroke DE of about 57% (Bitzer et al., 2016). The median stroke location error is 631 m (Mallick et al., 2015). In this work, we use the flash product provided by ENTLN. This product is based on the flash criteria





proposed by Liu and Heckman (2011), to cluster these strokes into flashes, in which two strokes are part of the same flash if they occur in a 0.7 s temporal window and in a 10 km spatial window.

After the end of operation of LIS on the TRMM satellite in 2015, a similar instrument was placed on the ISS for a 2 to 4 year mission starting in March 2017 covering latitudes between 54.3°N and 54.3°S (Blakeslee et al., 2014, 2020). LIS detects optical emissions from lightning with a frame integration time of 1.79 ms (Bitzer and Christian, 2015) with a spatial resolution of 4 km (Blakeslee et al., 2020). LIS assorts contiguous events into groups, and clusters groups into flashes with a temporal criteria of 330 ms and an spatial criteria of 5.5 km (Mach et al., 2007). Bitzer (2017) proposed a method to identify LCC

lightning flashes from the groups reported by TRMM-LIS. According to Bitzer (2017), optical emissions detected in five or more consecutive frames (time contiguous groups), that are in the same flash, can be classified as a LCC lightning flash. In this work, we use the method proposed by Bitzer (2017) to produce a climatology of LCC lightning flashes over Europe based on ISS-LIS lightning measurements between March 2017 and September 2020. We use 20 ms as the lower limit to classify a flash as a LCC flash in order to avoid introducing CG flashes without continuing currents (typical CG) into our climatology.

## 2.2   Forest fire databases

The data of fires in Spain is provided by the Spanish Ministerio de Agricultura, Pesca y Alimentación (López-Santalla and López-Garcia, 2019). The database includes fires for entire Spain between 1968 and 2015 and for all the region except for the region Castilla y León between 2016 and 2017. The causes of fires is provided when known. LIW are identified by the Spanish Ministerio de agricultura, Pesca y Alimentación using lightning data from the Spanish State Meteorological Agency

(AEMET). As the set of forest fires between 2016 and 2017 is not complete, we will exclude them from the analysis except for calculating the spatial flash density associated with LIW. The total number of fires in this database between 2009 and 2015 is 92187, among which 3162 are ignited by lightning (about 3.2%). In this study we do not include forest fires over the Canary Islands.

Data of fires in Portugal are provided by the Instituto da Conservaçao da Naturaleza e das florestas (Fernandes, 2015). The

database includes fires between 1980 and 2015. The cause of most fires is not provided. The total number of fires in this database between 2009 and 2015 is 163190. We describe in Section 2.4 a criterion to filter out some LIW from this data set, obtaining 359 LIW (about 0.22%).

Data of fires in the Mediterranean area in France are obtained from the Prométhée database (Délégation à la Protection de la Forêt Méditerranéenne, 2020). The database includes fires between 1973 and 2018. Cause of fires are provided. In this study,

we only include fires with reported geographical coordinates. The total number of fires in this database between 2009 and 2015, in which the geographical coordinates are provided, is 797, among which 36 are ignited by lightning (about 4.5%).

Hellenic Fire Brigade, collects and publish fire incidence data from 2000 onwards, after getting (in summer of 1998) the responsibility of fire extinction in Greece from Forest Service, including the number of fires and the corresponding are burned (Koutsias et al., 2013). During the last years, x and y coordinates of the fire ignition points are also recorded, unfortunately

without recording the fire cause. For three years during the period 2017-2019, these data were further checked for providing high positional accuracy and made available to be used here. The total number of fires in this database between 2017 and 2019





is 62690. We use the criterion described in Section 2.4 to filter out some LIW from this data set, obtaining 1999 LIW (about 3.2%).

## 2.3 Meteorological data and satellite measurements

We analyze the meteorological conditions of thunderstorms favoring fires or LCC lightning using meteorological data from the European Centre for Medium-Range Weather Forecasts (ECMWF) fifth generation reanalysis ERA5) (Hersbach et al., 2020) and the Cloud Top Height (CTH) product provided by EUMETSAT (Schmetz et al., 2002). In addition, for some selected cases we analyze the meteorological products provided by the Spanish State Meteorological Agency (AEMET) (Gutiérrez Núñez et al., 2018) for some selected cases.

ERA5 provides 1-hourly meteorological data using a 4D-var assimilation scheme at 139 pressure levels with an horizontal resolution of 0.25°. The product ERA5-Land provides meteorological data over land by replaying the land component of the ECMWF ERA5 climate reanalysis with an horizontal resolution of 0.1° (Poli et al., 2016). In this work, we analyzed the 1-hourly ERA5 meteorological data for cell and time step containing lightning flashes reported by WWLLN and ENTLN in the Iberian Peninsula and Mediterranean France for the period 2009-2015 and in Greece for the period 2017-2019. In particular,

we have included in the analysis ERA5 meteorological data for 133147 and 78701 cells/hours over, respectively, the Iberian Peninsula and Greece. In addition, we analyzed the ERA5 thermodynamical, dynamical and microphysical variables for 8351 cells/hours including lightning flashes reported by ISS-LIS over Europe between 2017-2019. The selected meteorological variables are the Convective Available Potential Energy (CAPE), the Cloud Base Height (CBH), the horizontal wind components, the hourly-accumulated precipitation, and vertical profiles between ground and 200 hPa pressure level of the temperature, the

relative humidity, the vertical velocity, the specific cloud ice water content, the specific cloud liquid water content, the specific rain water content and the specific snow water content.

The AEMET provides meteorological measurements under request. In this work, we selected some fire-igniting lightning flashes and some LCC lightning over Spain and collected radar echo top height measurements provided by AEMET (maximum height of the 12-dBZ echo). These heights correspond to the maximum heights populated by the precipitable particles.

We use the land cover map from 2015 produced by the European Space Agency (ESA) Climate Change Initiative (CCI) from the PROBA-Vegetation (PROBA-V) and Sentinel-3 OLCI (S3 OLCI) time series to classify lightning flashes over coniferous and mixed forests. This classification of lightning flashes ensures that only lightning flashes occurring over areas with vegetation that can be ignited are included in our analysis, as most of forest fires are due to the ignition of conifers or shrublands during the summer season (Pineda and Rigo, 2017). This map has a temporal resolution of one year and a horizontal resolution

of 300 m. We consider that a lightning flash is taking place over coniferous or mixed forests if there is at least one 300 m × 300 m grid cell with conifer or mixed trees 10 km around the flash position. In particular, we consider coniferous and mixed forests that including the label "needleleaved" in the D3.3.12-v1.3_PUGS_ICDR_LC_v2.1.x_PRODUCTS_v1.3 product. We use elevation data from the NASA Shuttle Radar Topographic Mission Farr et al. (2007) to get the elevation of LIW.

The CTH product provided by EUMETSAT is based on measurements of the Meteosat Second Generation (MSG) satellites.

The geostationary orbit of MSG satellites is centered at 0°E, 0°N, reporting data at the rate of one Earth full-disk scan every



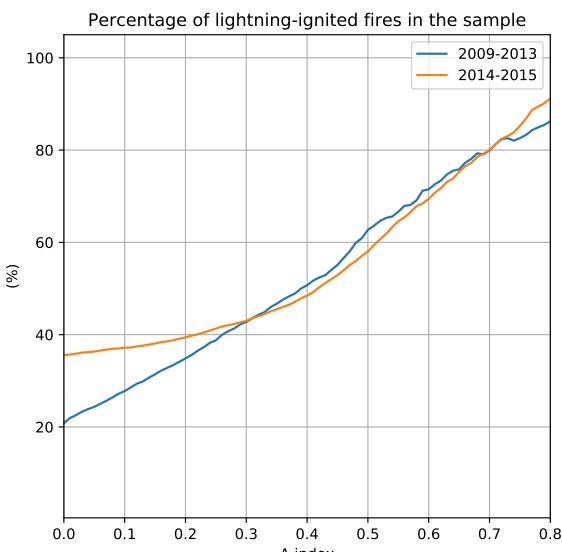

**Figure 1.** Percentage of LIW with different $A$ index values as threshold in Spain and France during the period 2009-2015.

15 min. We investigate the CTH of fire-igniting thunderstorms because it is closely related with the level of convection in thunderstorms and the occurrence of lightning (Price and Rind, 1992). The CTH product is calculated by EUMETSAT from data acquired by the Spinning Enhanced Visible and InfraRed Imager (SEVIRI) instrument onboard the MSG satellites with an horizontal resolution of 4 km at the center of the orbit and a vertical resolution of 320 m, reaching a maximum altitude of

16 km. The CTH product is provided every 15 min since June 2011, with a lower rate between 2009 and May 2011 (Schmetz et al., 2002). In this work, we collect all the possible CTH values for the lightning flashes reported by WWLLN and ENTLN in the Iberian Peninsula (and Mediterranean France) between 2012-2015. We also collect the CTH values for all the lightning flashes reported by ENTLN in Greece for the period 2017-2019.

### 2.4    Search of lightning-candidates for the fires

We search the most probable CG lightning candidate for each fire using the proximity index $A$ proposed by Larjavaara et al. (2005b),

$$A = \left(1 - \frac{T}{T_{max}}\right) \times \left(1 - \frac{D}{D_{max}}\right). \tag{1}$$

The proximity index $A$ combines the distance between each fire and each lightning discharge ($D$) with the delay between them ($T$), also known as holdover. Parameters ($T_{max}$) and ($D_{max}$) correspond to the maximum holdover and distance between

a fire and a lightning discharge to consider the later as the cause of ignition. We set $T_{max}$ = 14 days and $D_{max}$ = 10 km





(Larjavaara et al., 2005b). In the case of no CG lightning discharges taking place within the proposed spatio-temporal window with respect to a fire, we consider that the fire was not ignited by lightning. However, the proximity index can be greater than 0 for fires that were not ignited by lightning.

In the case of the database of fires taking place in Spain and Mediterranean France, the cause of ignition is provided.
Therefore, we can easily discard the fires with $A > 0$ that were not ignited by lightning. However, the database of fires in Portugal and Greece do not provide the cause of ignition, and the proximity index can be greater than zero if a lightning discharge preceded a fire even if the discharge did not cause the ignition. In order to discard the fires of Portugal and Greece that were not ignited by lightning, we estimate the threshold value of the $A$ index that ensures that at least 80% of the selected fires are ignited by lightning. To this end, we use the Spanish and French databases of fires. We plot in Fig. 1 the percentage
of fires in Spain and France ignited by lightning using different values of $A$ as threshold. According to this figure, setting the minimum value of $A$ as 0.7 ensures that 80% of the selected fires are produced by lightning. Therefore, in our analysis of LIW over Portugal and Greece we only consider fires with $A{\geq}0.7$. This approach can exclude a significant number of LIW from the analysis. However, it ensures that the sample is not influenced by fires that were not ignited by lightning. Following this approach, we find 359 (0.22% of the total) and 1999 (3.2% of the total) LIW with $A{\geq}0.7$ in Portugal and Greece, respectively.
We have obtained a lower percentage of LIW in Portugal with respect to all fires than in Greece, Spain and Mediterranean due to the low DE of WWLLN. Another interesting feature of Fig. 1 is that the percentage of LIW in the sample with $A$ values lower than 0.3 is higher in the period 2014-2015 than in 2009-2013. This difference can be explained by the higher DE of ENTLN than WWLLN in the Iberian Peninsula.

As explained in Section 2.3, we will limit our analysis to LIW occurring between May and September and in coniferous and
mixed forest.

### 2.5 Classification of LCC lightning flashes

The standard ISS LIS product includes lightning data at event, group and flash level (Mach et al., 2007). Groups are clustered into flashes if they occur within 330 ms and 5.5 km (Mach et al., 2007). In this work, we use group and flash data. We apply the method to classify lightning flashes reported by ISS-LIS as LCC lightning flashes proposed by Bitzer (2017) for TRMM
LIS. We consider a flash as a LCC lightning flash if its optical emissions are detected in ten or more consecutive frames, which entails that the detected optical signal is continuous during about 20 ms. In addition, we have classified flashes with optical emissions detected in five or more consecutive frames (optical signal is continuous during about 10 ms (Bitzer, 2017)) to investigate the possible relationships between the duration of the current and the meteorological conditions.

### 2.6 Analysis of meteorological conditions

We compare the meteorological variables of fire-igniting lightning flashes to typical CG lightning flashes over coniferous and mixed forest between May and September to identify the characteristics of fire-igniting flashes. In addition, we compare the meteorological conditions associated with LCC lightning flashes reported by ISS-LIS over land in Europe between May and September 2017-2020 to the meteorological conditions of typical lightning flashes to search possible relationships between





meteorology and the occurrence of LCC lightning flashes. The comparison of the meteorological variables is performed as
follows:

1. We collect the 1-hourly values of the meteorological variables of every ERA5 grid cell containing lightning flashes.
For forecasting purposes, we collect the maximum value 3 hours before the discharge, as the value of CAPE during the
passage of a thunderstorm is significantly lower than the value of CAPE during the development of the storm. For every
lightning discharge, we collect the CTH value provided by EUMETSAT.

2. We define four samples: A first sample containing the meteorological variables of all the selected CG lightning discharges
(*CG lightning climatology*), a second sample formed by the meteorological variables of all the fire-igniting lightning
candidates (*fire-igniting lightning climatology*), a third sample containing the meteorological variables of all the LCC
lightning flashes (*LCC flashes climatology*) and a fourth sample composed by the meteorological variables of all the
lightning flashes reported by ISS-LIS without a continuing phase (*Typical lightning flashes*).

3. We calculate the median values of each sample and compare them.

4. We perform a Kruskal-Wallis H-test between both samples to check if the median value differs significantly across
samples and calculate the corresponding $p-value$. The $p-value$ is the level level of marginal significance within the
statistical hypothesis of equal median values, indicating the probability of equal median for both samples. If the $p-value$
is lower than 0.05 (less than 5% probability of equal median) we reject the hypothesis of equal medians and consider
that both samples are statistically different (Kruskal and Wallis, 1952). It is important to mention that different medians
does not necessary imply that the analyzed variable would be a good predictor of LIW.

## 3 Results

### 3.1 Lightning-candidates for fires

The number of *LIW* over coniferous and mixed forest included in our analysis over Spain, Portugal, France and Greece are
3162, 359, 36 and 1999 (Table 1), respectively. In Fig. 2 we plot two maps showing the Iberian Peninsula and Greece, the
position of coniferous/mixed forests and the position of all the LIW included in this study.

From this point, we refer to the contiguous territory including Spain, Portugal and Mediterranean France and the Balearic
islands as the Iberian Peninsula. Fig. 3(a) and 4(a) show the monthly total number of LIW between May and September in the
Iberian Peninsula and in Greece, respectively. In the Iberian Peninsula, the occurrence of LIW reaches its peak in July, while
in Greece the peak occurs in June and a secondary peak in September.

Fig. 3(b) and 4(b) show the frequency distribution of the lightning peak current of all the lightning taking place over conifer-
ous and mixed forests and fire-igniting lightning during May and September for, respectively, the Iberian Peninsula and Greece.
We do not include peak currents during 2009 and 2013, as the WWLLN lightning data does not include information about the
polarity and the peak current of the flash. According to these figures, there is a slight shift to higher peaks with a positive



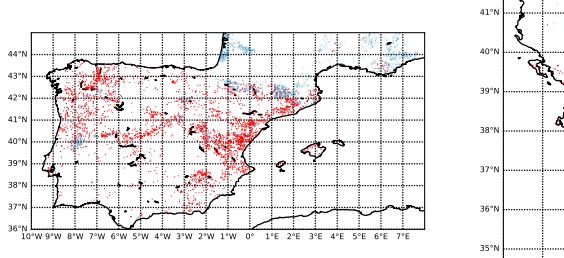
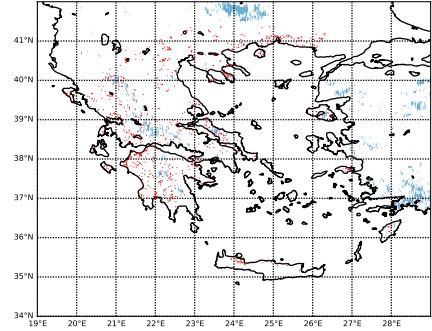

**Figure 2.** Coniferous and mixed forest (blue area) and LIW (red dots) included in this study. Left panel shows LIW over the Iberian Peninsula between 2009 and 2015, while right panel shows LIW over Greece between 2017 and 2019. We have degraded the resolution of the vegetation map in this plot from 300 m to 4.8 km.

polarity. The lack of positive lightning flashes reported by ENTLN with peak currents below 20 kA are a consequence of the data set, that do not include positive CG flashes with peak currents below nearly 20 kA. Positive flashes with peak currents below nearly 20 kA are classified as IC (Cummins and Murphy, 2009).

Fig. 3(c-d) and 4(c-d) show the distance and holdover between when the reported fire ignition and the lightning candidates for the Iberian Peninsula and Greece, respectively. In the Iberian Peninsula, the 25th, 50th and 75th percentiles of the distance

are respectively 1.30 km, 2.71 km and 5.39 km, while in Greece they are 0.56 km, 0.93 km and 1.48 km. The distance between fires and fire-igniting flashes is significantly lower in the case of Greece, possibly due to improvement in the detection efficiency of ENTLN between the periods 2014-2015 and 2017-2019. The 25th, 50th and 75th percentiles of the holdovers in the Iberian Peninsula are, respectively, 0.48 hours, 8.13 hours and 29.23 hours, while they are 14.79 hours, 29.43 hours and 48.78 hours in Greece. Differences in the holdover between the Iberian Peninsula and Greece can be due to differences

in the detection of fires by local authorities. Fig. 3(d) and 4(d) suggest a diurnal cycle in the holdover that can be due to different conditions for arrival during day, similar as obtained by Pineda and Rigo (2017) for LIW ober Catalonia. At noon, meteorological conditions can favor a rapid arrival after ignition (high temperature and low humidity). Ignitions separated from noon occur under meteorological conditions that do not favor rapid arrival.

We have found that the mean elevation of LIW in the Iberian Peninsula is 775 m, while in the case of Greece the mean

elevation is 319 m.

### 3.2  Meteorological conditions of fire-igniting lightning

We investigate the meteorological conditions of LIW using 1-hourly ERA5 reanalysis data. We compare several meteorological variables of *fire-igniting lightning* with *typical CG lightning* taking place over coniferous and mixed forest between May and September to find the meteorological conditions that favor the ignition of fires by lightning.



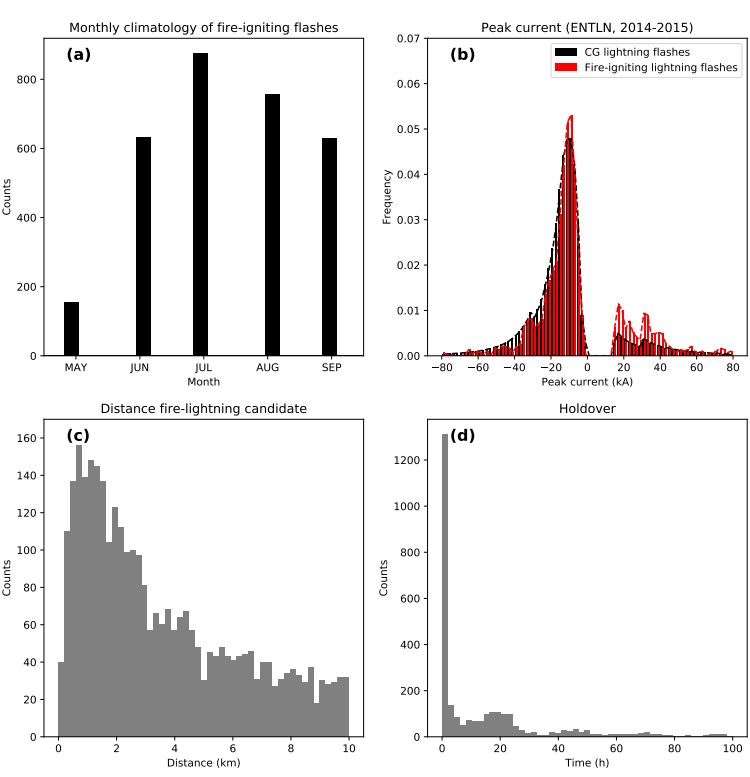

**Figure 3.** Analysis of fire-igniting flashes in the Iberian Peninsula between 2009 and 2015: (a) Monthly distribution of the occurrence of fire-igniting lightning flashes, (b) Frequency distribution of the peak currents of all CG lightning taking place over coniferous and mixed forest during May and September (black) and of all fire-igniting lightning flashes (red) provided by ENTLN between 2014 and 2015. (c) Distribution of the distance between the reported position of ignition and the lightning candidate. (d) Distribution of the holdover (difference between the time of detection and the time of the lightning candidate).

Fig. 5(a) and Fig. 6(a) show the frequency distributions of CAPE for the CG lightning and the fire-igniting lightning climatologies for the Iberian Peninsula and Greece, respectively. We have selected the maximum value of CAPE 3 hours before the occurrence of lightning. Comparison of both distributions indicates that low values of CAPE are more frequent in fire-igniting flashes than in typical CG lightning flashes over the Iberian Peninsula. However, we find an opposite tendency over Greece. This is not surprising, as according to Nauslar et al. (2013), lower tropospheric thunderstorms indices, such as CAPE, are not
enough to forecast the potential of dry thunderstorms. Nevertheless, the p-values lower than 0.05 suggest that local differences in the distributions of CAPE plotted in Fig. 5(a) and Fig. 6(a) are statistically significant.

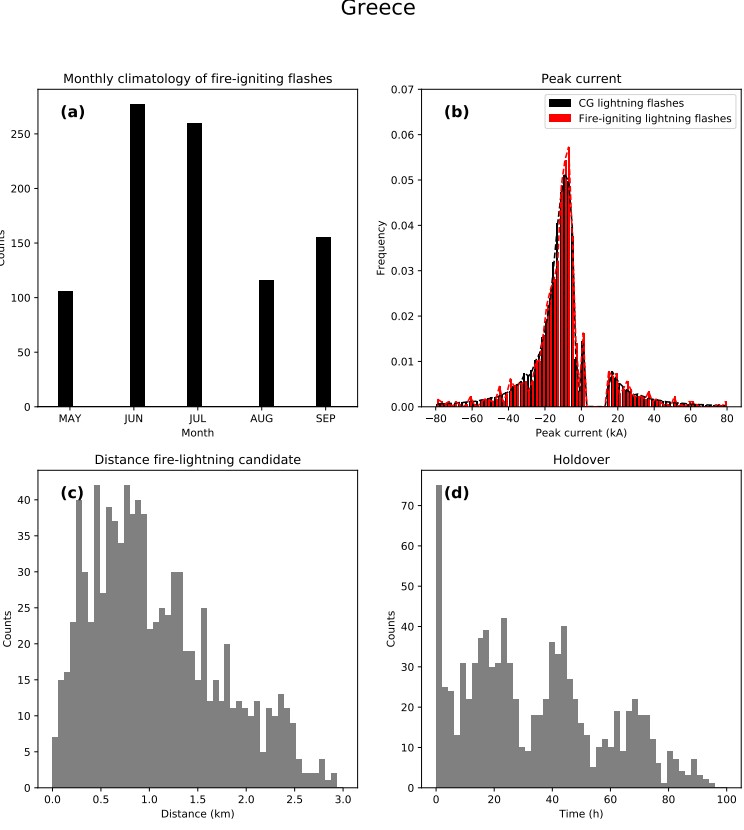

**Figure 4.** Analysis of fire-igniting flashes in Greece between 2017 and 2019: (a) Monthly distribution of the occurrence of LIW, (b) Frequency distribution of the peak currents of all CG lightning taking place over coniferous and mixed forest during May and September (black) and of all fire-igniting lightning flashes (red) detected with ENTLN. (c) Distribution of the distance between the reported position of ignition and the lightning candidate. (d) Distribution of the holdover.

Fig. 5(b) and Fig. 6(b) show the frequency distribution of the hourly accumulated precipitation for the CG lightning and the fire-igniting lightning climatologies. In accordance with previous studies over the US (e.g., Colson, 1960; Hall, 2007; Pineda and Rigo, 2017), our results also suggest that thunderstorms with a low precipitation rate are associated with a higher

probability of lightning-ignited fire production over Europe. In the Iberian Peninsula, the 25th, 50th and 75th percentiles of the hourly accumulated precipitation for LIW are respectively 0.10 mm, 0.63 mm and 2.01 mm, while for Greece the percentiles are respectively 0.13 mm, 0.88 mm and 2.76 mm. Again, the p-values suggest that differences in the distributions of the accumulated precipitation plotted in Fig. 5(b) and Fig. 6(b) are statistically significant.

We plot in Fig. 5(c) and Fig. 6(c) the comparison of the frequency distribution of the horizontal wind at the surface for the

CG lightning and the fire-igniting lightning climatologies. Differences in the median values suggests that stronger winds favor





the arrival of fires (distribution shifted to the right) in the Iberian Peninsula and Greece. The p-values suggest that differences in the distributions of the horizontal winds plotted in Fig. 5(c) and Fig. 6(c) are statistically significant.

Fig. 5(d-f) and 6(d-f) show the frequency distribution of, respectively, the Relative Humidity (RH) at 850 hPa level, the air temperature at 2 m altitude and the air temperature at 850 hPa level pressure for the CG lightning and the fire-igniting
lightning climatologies. The lower RH and the higher temperature at 850 hPa and 2 m altitude are characteristic conditions of dry thunderstorms over the US (Rorig et al., 2007; Dowdy and Mills, 2012), Australia (Bates et al., 2017; Dowdy, 2020). These conditions favor the evaporation of precipitable water before reaching the ground (Nauslar et al., 2013) increasing the probability of fire-ignition, survival and arrival in the Iberian Peninsula and Greece. Differences in the distributions of RH and temperatures plotted in Fig. 5(d-f) and 6(d-f) are statistically significant, as suggested by p-values lower than 0.05. However,
Fig.6(d, f) suggest that the Relative Humidity (RH) at 850 hPa level and a the air temperature at 850 hPa level pressure would not be good predictors for LIW in Greece, as the plotted distributions of fire-igniting lightning and CG lightning are not substantially different. Differences in RH and temperature at 850 hPa between Greece and the Iberian Peninsula could be due to topographic effects. LIW in the Iberian Peninsula tend to occur at higher elevation than in Greece (775 m and 319 m, respectively). Therefore, meteorological conditions at 850 hPa would have a greater influence on ignition, survival and arrival
of fires in the Iberian Peninsula than in Greece.

We have also calculated the median value of the variables plotted in 6 using $A \geq 0.8$ and $A \geq 0.85$. We have obtained the same results as using $A \geq 0.7$ in terms of difference between meteorological variables of fire-igniting lightning and typical lightning. However, the p-values are significantly higher, as the total number of fire-igniting lightning included in the sample decreases significantly when increasing the minimum value of $A$.

The temperature difference between vertical levels can be used to investigate the instability of the atmosphere. Let us now analyze the temperature differences between 2 m altitude and 850 hPa pressure level and between 850 hPa and 450 hPa pressure levels. We have found that the median temperature differences between 2 m altitude and 850 hPa pressure level for CG lightning in the Iberian Peninsula and Greece are, respectively, 5.6 K and 7.5 K. In the case of fire-igniting lightning, the median temperature differences between 2 m altitude and 850 hPa pressure level are 6.3 K and 8.5 K, respectively. The median
temperature differences between 850 hPa and 450 hPa pressure levels for CG lightning in the Iberian Peninsula and Greece are 33.5 K and 33.1 K, respectively, while in the case of fire-igniting flashes the median temperature differences between 850 hPa and 450 hPa pressure level for the Iberian Peninsula and Greece are, respectively, 35.4 K and 33.2 K. These results indicate that the instability of the atmosphere is higher in fire-igniting lightning flashes than in typical CG lightning flashes. However, we have found some differences between the Iberian Peninsula and Greece. The highest difference in the temperature of the
atmosphere between typical CG lightning and fire-igniting lightning flashes in Greece takes place in the lower troposphere (between 2 m altitude and 850 hPa pressure level), while the highest difference in the instability of the atmosphere between typical CG lightning and fire-igniting lightning flashes in the Iberian Peninsula occurs at higher levels (between 850 hPa and 450 hPa pressure levels). These differences can be explained by the different mean elevation of LIW in the Iberian Peninsula and in Greece.





**Table 2.** Median value of the distributions of fire-igniting lightning plotted in Fig. 5-7 for each of the climate zones of the Iberian Peninsula and Greece. We include the median value of the distributions of CG lightning flashes in brackets and the p-values (zero when lower than the machine precision). We also include in this table the median value of the meteorological variables of LCC(>20 ms)-lightning reported by ISS-LIS taking place in high-based clouds (Fig. 18) and the CBH of all LCC(>20 ms)-lightning (Fig. 21), including the median value of the distributions of typical lightning flashes observed by ISS-LIS in brackets.

| Region | CAPE (J kg$^{-1}$) | Precipitation (mm) | Wind (m s$^{-1}$) | RH (%) | Temperature (K) | Temperature 850 hPa (K) | CBH (m) |
|---|---|---|---|---|---|---|---|
| *Forest fires* | | | | | | | |
| **Iberian Peninsula** | 361 (433) | 0.63 (1.70) | 2.04 (1.79) | 57.4 (74.1) | 297 (293) | 291 (287) | 1739 (1157) |
| p-value | < 0.001 | < 0.001 | < 0.001 | < 0.001 | < 0.001 | < 0.001 | < 0.001 |
| *Atlantic* | 315 | 0.77 | 1.81 | 62.3 | 295 | 290 | 1747 |
| *Mediterranean* | 249 | 0.71 | 2.24 | 56.9 | 297 | 291 | 1656 |
| *Central* | 99 | 0.38 | 2.04 | 49.3 | 297 | 292 | 1846 |
| *Pyrenees* | 335 | 0.45 | 1.9 | 60.5 | 296 | 290 | 1723 |
| **Greece** | 666 (479) | 0.88 (1.16) | 1.69 (1.44) | 69.8 (72.1) | 297 (295) | 288 (288) | 1694 (1350) |
| p-value | < 0.001 | < 0.001 | < 0.001 | < 0.001 | < 0.001 | < 0.001 | < 0.001 |
| *LCC(20 ms)-lightning flashes* | | | | | | | |
| **Europe (land)** | 288 (354) | 0.30 (1.21) | - | 64.1 (71.3) | 296 (295) | 289 (288) | 1361 (1386) |
| p-value | 0.16 | < 0.001 | - | < 0.001 | < 0.001 | < 0.001 | 0.7 |

Fig. 7 shows the frequency distribution of the CBH for lightning flashes over coniferous and mixed forest and for all the fire-igniting flashes, indicating that high-based clouds favor the inception of LIW. High-based clouds favor the evaporation of precipitation before reaching the surface, increasing the probability of survival and arrival of LIW. High-based clouds and low content of moisture in the low- or mid-level are the typical meteorological conditions of dry thunderstorms in the western United States, as demonstrated by Wallmann (2004) and Nauslar et al. (2013). Our results suggest, for the first time, that high-based clouds favor the occurrence and arrival of LIW in the Iberian Peninsula and Greece. The p-values suggest that differences in the distributions of CBH plotted in Fig. 7 are statistically significant.

### 3.2.1 Regional patterns over the Iberian Peninsula

The inland area of the Iberian Peninsula is influenced by a complex spatial climate pattern. Climate in the Atlantic coast is characterized by a high accumulation of precipitation due to the frequent Atlantic storms. The Mediterranean coast has a climate characterized by dry conditions during the summer, while the inland central part of the Iberian Peninsula is influenced by a continental climate with low precipitation and extreme temperatures during winter and summer. Finally, some areas, as the Pyrenees, are influenced by a highland climate. We show in Table 2 the median value of the distributions plotted in Fig. 5 for each of the climate zones of the Iberian Peninsula. In all regions the median values of 1-hour precipitation and relative humidity are lower than the median values for all CG lightning flashes, while the median values of the wind, the temperatures at 2 m and 850 hPa and the CBH are higher than the median values for all CG lightning flashes. The median value of CAPE for fire-igniting lightning in the Iberian Peninsula is lower than the climatological median.

### 3.2.2 Meteorological conditions in the upper troposphere

Let us now investigate the meteorological conditions of thunderstorms at altitudes above the 850 hPa pressure level. Hydrometeors (in the shape of water or ice) can play an important role for the lightning activity (Deierling et al., 2005; Finney et al.,





2014) and for the precipitation rate (Tao et al., 2012). We plot in Fig. 8 and Fig. 9 the vertical profiles of the specific cloud ice water content and the specific rain water content for the CG lightning and the fire-igniting lightning climatologies during May and September in the Iberian Peninsula and Greece, respectively. We plot in the second column of Fig. 8 and 9 the vertical profile of the p-values to indicate the pressure levels where differences are statistically significant.

The total cloud ice is lower for the case of fire-igniting lightning than for the CG lightning climatology (Fig. 8(a) and
Fig. 9(a), except at altitudes above 450 hPa pressure levels in the case of Greece. The lower content of ice particles in fire-igniting lightning suggests that fire-producing thunderstorms have a lower content of moisture than the climatological median. The low rain content in the lower troposphere (altitude below 700 hPa pressure level) for fire-igniting lightning (Fig. 8(b) and Fig. 9(b)) suggests that fire-igniting lightning flashes occur in dry thunderstorms (Nauslar et al., 2013). The red line in Fig. 8(b) disappears when the value is lower than $10^{-8}$ below about 750 hPa pressure level, indicating that the specific rain
water content is negligible. The only pressure levels where differences are not statistically significant (p-value greater than 0.05) are the uppermost levels plotted in each panel of Fig. 9.

Finally, we plot in Fig. 10 and Fig. 11 the vertical profile of the vertical velocity for the CG lightning and the fire-igniting lightning climatologies in the Iberian Peninsula and in Greece, respectively. The vertical velocity is the speed of air motion in the upward or downward direction, with negative values for upward motion and positive values for downward motion. It is
important to mention that the median value that we have obtained is lower than zero at all pressure levels, which means that the vertical velocities plotted in Fig. 10 and Fig. 11 correspond to updrafts. Fig. 10 indicates that fire-igniting lightning in the Iberian Peninsula tends to occur in areas with weaker vertical motion than in the case of typical CG lightning (p-value lower than 0.05 at every level). However, we do not find the same trend in Greece. Fig. 11 shows that vertical velocity is lower for fire-igniting lightning than for CG lightning between 300 hPa and 450 hPa, higher for altitudes below 800 hPa pressure level
and statistically similar in the rest of the vertical levels.

### 3.2.3  Cloud Top Height values

Let us now compare the CTH values associated with fire-igniting lightning and with typical lightning using the CTH product provided by EUMETSAT and based on measurements of the MSG satellites. Thunderstorm with high CTH values are related with high level of convection and lightning activity (Price and Rind, 1992). We have collected the value of the CTH for all CG
lightning flashes and fire-igniting lightning flashes for the periods between 2012 and 2015 (Iberian Peninsula) and between 2017 and 2019 (Greece) when it is provided in a temporal window of $\pm 15$ minutes around the occurrence of the discharge and when the reported quality of the estimation of CTH is not poor. As an example, we plot in the supplement the CTH map derived from MSG satellite for one fire-igniting thunderstorm taking place in the Mediterranean coast of the Iberian Peninsula in June 15, 2015 at 14:15 UTC.

We plot the frequency distribution of CTH values in Fig. 12 for both the CG lightning and the fire-igniting lightning climatologies in the Iberian Peninsula (upper panel) and in Greece (lower panel). We obtain a median CTH value of 11.3 km (10.3 km) for fire-igniting lightning and a median value of 11.3 km (10.6 km) for typical CG lightning flashes in the Iberian Peninsula (Greece). There are not significant differences in the distribution of CTH values between both climatologies over



the Iberian Peninsula (p-value>0.05). However, Fig. 12 shows a tendency to higher CTH values for fire-igniting lightning in
Greece. The frequency distribution of CTH is flatter for the case of fire-igniting lightning flashes than for the case of CG
lightning flashes. The high total number of fire-producing thunderstorms over Greece with CTH above 12 km showed (lower
left panel of Fig. 12) coincides with the high content of ice water at 200 hPa plotted in Fig. 9.

### 3.3 Lightning flash frequency and density in fire-producing thunderstorms

Lightning flash frequency and density is used as a proxy variable to forecast or parameterize the inception of lightning ignited
fires. For example, Krause et al. (2014) parameterized the inception of LIW in a global model using the flash frequency, while
Schultz et al. (2019) found that the majority of flash densities were less than 0.41 flashes per $km^{-2}$ in lightning-ignited fire
days in the United States. In this section, we analyze the *temporal evolution* and the *spatial characteristics* of fire-producing
thunderstorms in Spain between 2014 and 2015. We assume that the fire-producing thunderstorm is the one that contains
the lightning-candidate. However, identifying the thunderstorm containing a flash reported by a lightning network is not a
trivial problem. Hutchins et al. (2014) proposed the use of the ST-DBSCAN algorithm to cluster lightning flashes reported
by WWLLN into thunderstorms. The ST-DBSCAN algorithm defines a cluster of flashes as the group of flashes that are
neighbors according to selected spatial and temporal thresholds. Hutchins et al. (2014) reported that ST-DBSCAN produces
a reliable daily global number of thunderstorms ($660 \pm 70$ on any given moment) by using $0.12°$ in latitude and longitude
as spatial parameter and 18 minutes as temporal thresholds to cluster WWLLN flashes into thunderstorms. In this work, we
cluster lightning flashes reported by ENTLN in the region of interest by searching adequate clustering thresholds by using
radar measurements of thunderstorms.

Radar echo top measurements at 12 dB provide the maximum height of precipitable particles (Gutiérrez Núñez et al.,
2018). Areas where the height of precipitable particles is high are related to areas with strong updrafts that can favor the
production of thunderclouds. We use radar echo top maps to identify the position of thunderstorms before clustering flashes
into thunderstorms. Fig. 13 (left column) show two echo top radar images around two fire-ignited lightning taking place in the
Mediterranean coast of the Iberian Peninsula on June 17, 2015 at 18:00 UTC and on June 15, 2015 at 14:15 UTC, respectively.
Fig. 13(right column) show all the lightning flashes reported by ENTLN in the Eastern coast of the Iberian Peninsula during
30 minutes on June 17, 2014 at 18:00 and on June 15, 2015 at 14:15 UTC. Different colors indicate the clusters after applying
the clustering algorithm ST-DBSCAN with the spatial and temporal parameters $0.12°$ and 15 minutes, respectively. Comparison
of both columns in Fig. 13 shows that the clustering parameters $0.12°$ and 15 minutes, can be used to identify thunderstorms
from lightning measurements in the region of interest. Hutchins et al. (2014) used 18 minutes for WWLLN global lightning
data. However, we found that using 18 minutes for ENTLN lightning data over the Iberian Peninsula would cluster all the
lightning flashes shown in Fig. 13(right panel) into the same thunderstorm.

Following this approach to cluster lightning flashes into thunderstorms, we have clustered the lightning flashes taking place
in the Iberian Peninsula between 2014 and 2015 during some days with LIW. Given that the clustering process requires high
computational resources, we restrict our analysis to the period between 2014 and 2015. In order to select the cases of study,
we have only included fire-igniting thunderstorms containing a lightning candidate with a proximity index ($A$) greater than





0.95 with respect to the fire. Following these criteria, we have selected 77 cases. Fig. 14 shows the cumulative number of flashes occurring in the selected cases 1400 minutes around the ignition of LIW. *Ignitions tend to occur at the moment when*

*the lightning activity just starts to increase from the regime with the lowest lightning activity to the regimen with the highest lightning activity*. The regime with a higher lightning activity coincides with a higher precipitation rate (Soriano et al., 2001) that contributes to inhibit the possibility of fire ignition. We have found no relationships between the cumulative number of different type and polarity of lightning discharges and the time of ignition.

Several authors have investigated the spatial density of CG flashes 24 h prior to the ignition as a proxy for the occurrence

of fires in the US depending on the 100-h fuel moisture information (e.g., Latham, 1989; Hardy et al.; Schultz et al., 2019). They have found a that the threshold value of the CG spatial flash density to ignite a fire and allow it to reach the arrival phase would strongly depend on the lightning detection system. In this work, we have calculated the CG spatial flash density 5 km around and 24 h before the ignition of all the LIW in Spain during 2016 and 2017 using CG flashes reported by ENTLN. The DE of ENTLN can have increased during recent years. We have calculated the CG spatial flash density for the period between

2016 and 2017 for the case of utility, as 2016-2017 are the most recent years in our fire database in the Iberian Peninsula. Therefore, is the most recent . We obtain a median value of 0.44 flashes per $km^{-2}$, a 25th percentile of 0.26 flashes per $km^{-2}$, and a 75th percentile of 0.69 flashes per $km^{-2}$. In addition to meteorological parameters, these values could be used in regional fire forecasting or atmospheric models to quantify the risk of ignition by thunderstorms, specially if radar measurements of precipitation rate are not available.

### 3.4 Long-Continuing-Current (LCC) lightning

In this section, we investigate the climatology of LCC lightning flashes over Europe derived from ISS-LIS data between March 2017 and September 2020 and the meteorological and cloud conditions under which they take place. This approach will help us to identify possible relationships between LCC lightning and fire-igniting lightning flashes.

#### 3.4.1 LCC lightning distribution over Europe

In order to perform this analysis, we compare three samples of lightning data. The first sample is formed by all lightning flashes without a long continuous phase of 20 ms or more reported by ISS-LIS over land in Europe between May-September 2017-2020, here referred as "Typical lightning flashes". The second and third samples are composed by all the LCC lightning flashes reported by ISS-LIS over land in southern Europe with a continuous phase duration equal or greater than 20 ms and 10 ms, respectively. We refer to these climatologies as, respectively, LCC(>20 ms) and LCC(>10 ms). The total number of

LCC(>20 ms) and LCC(>10 ms)-lightning flashes reported by ISS-LIS over land in Europe between May-September 2017-2020 are 1227 and 9211, respectively, while the total number of typical lightning flashes is 148482. Therefore, about 6.2% lightning flashes reported by ISS-LIS over Europe are LCC(>10 ms), while only about 0.8% lightning flashes are LCC(>20 ms). We plot the obtained geographical distribution of LCC(>20 ms)-lightning flashes in Fig. 15 together with the geographical distribution of lightning flashes reported by ISS-LIS. According to this figure, LCC(>20 ms)-lightning flashes tend to occur

over the oceans and over coastal regions, even when the maximum occurrence rate of total lightning flashes is produced over





land. Lightning flashes tend to be more energetic over ocean than over land (Said et al., 2013; Holzworth et al., 2019), which is in agreement with the different ratio of LCC(>20 ms)-lightning flashes to total lightning flashes plotted in the top-right panel of Fig. 15.

Comparison of LIW maps (Fig. 2) and LCC(>20 ms)-lightning maps (Fig. 15) in the Iberian Peninsula and Greece is difficult
as a consequence of the low total number of reported LCC(>20 ms)-lightning flashes by ISS-LIS. However, it is interesting to highlight the high occurrence of LCC(>20 ms) flashes and fire-igniting lightning in the Mediterranean coast of the Iberian Peninsula. In addition, we have obtained a high occurrence of LIW and LCC(>20 ms)-lightning flashes over the Greek coastal regions and over some parts of the Pindus Mountains.

Fig. 16 shows the monthly occurrence of typical lightning and LCC(>20 ms)-lightning flashes over land in Europe. The
peak in the occurrence rate of LCC(>20 ms)-lightning flashes is reached during the summer season, while the peak in the ratio of LCC(>20 ms)-lightning to typical lightning flashes occurs during the winter. The total number of flashes during winter is lower than during summer, causing that the ratio of LCC(>20 ms)-lightning to typical lightning flashes oscillates more during winter than during summer. Winter thunderstorms are characterized by weak updrafts, which suggests that weaker convection in thunderstorms favors the occurrence of LCC(>20 ms)-lightning flashes, as proposed by Bitzer (2017).

**3.4.2   Meteorological conditions for LCC lightning**

With the purpose of finding possible relationships between LIW and LCC(>20 ms)-lightning flashes, we analyze the meteorological conditions associated with LCC(>20 ms)-lightning flashes reported by ISS-LIS over land in Europe between May and September 2017-2020. We exclude winter and autumn seasons (January-April and October-December) LCC(>20 ms)-lightning data from our analysis because the total number of LIW during those seasons is negligible.

We show in the supplement the frequency distribution of CAPE, the accumulated precipitation, the relative humidity at 850 hPa pressure level and the temperature at 2 m altitude and at 850 hPa pressure level for typical lightning and LCC(>20 ms) lightning flashes. We also show the analysis of the vertical content of moisture in the supplement, that suggest that LCC(>20 ms) flashes tend to occur in thunderstorms with lower content of moisture than the climatological median. We have added these values to Table 2. We have not found any evident link between these meteorological conditions at the lower-troposphere and
the occurrence of LCC(>20 ms) lightning flashes.

However, two of the meteorological parameters that are relevant for the occurrence of fire-igniting lightning (Section 2.3) are also related with the occurrence of LCC(>20 ms)-lightning flashes: the vertical velocity and the CBH. The median vertical profiles of the vertical velocity for typical and LCC(>20 ms)-lightning plotted in Fig. 17 indicate that *LCC(>20 ms)-lightning flashes tend to occur under weaker convection than typical lightning flashes*. The p-value is below 0.05 for all the pressure
levels at altitudes below 400 hPa, indicating that differences in medians of both distributions are statistically significant.

We plot in Fig. 18 the frequency distribution of the CBH for typical lightning and LCC(>20 ms)-lightning. The median value of the CBH associated with LCC(>20 ms)-lightning flashes is similar as the value of CBH for typical lightning. The distribution of CBH for LCC(>20 ms)-lightning has two peaks, one of them in about 500 m and another in about 1250 m, while the distribution of CBH for typical lightning peaks in nearly 1000 m. The frequency distribution of CBH for LCC(>20 ms)-





lightning is higher than the frequency distribution for typical lightning in low-based clouds (CBH lower than 500 m) as well
as in high-based thunderstorms (CBH>2000 m). The p-value is below 0.05, indicating that differences in medians of the
showed distributions are statistically significant. Probably, LCC(>20 ms)-lightning flashes in low-based clouds do not produce
a significant amount of LIW because they are accompanied by high precipitation rates that do not favor the survival and arrival
phases of fires. However, LCC(>20 ms)-lightning occurring in high-based clouds could be the main precursors of LIW. We

will explore this possible relationship between LCC(>20 ms)-lightning and LIW in Section 3.4.3.

We plot as red dots the LCC(>20 ms)-lightning flashes in thunderstorm with CBH>2000 m in the first panel of Fig.15.
The geographical distribution of LCC(>20 ms)-lightning flashes in thunderstorm with CBH>2000 m suggests that they tend
to occur over mountains. However, a climatology covering a larger number of years is needed to explain the geographical
distribution of LCC(>20 ms)-lightning flashes with CBH>2000 m.

Fig. 19 shows the frequency distribution of CTH values of typical and LCC(>20 ms)-lightning flashes reported by ISS-LIS
over Europe between 2017 and 2019. The median CTH value of LCC(>20 ms)-lightning flashes is slightly below the median
CTH value of typical lightning flashes (11.1 km). These results suggest that LCC(>20 ms)-lightning flashes tend to occur in
thunderstorms with weaker convection than the median, as suggested by Bitzer (2017) for LCC(>10 ms)-lightning. The p-value
is below 0.05, indicating that differences in medians of the showed distributions are statistically significant.

### 3.4.3   Lightning flash frequency in LCC(>20 ms)-producing thunderstorms

For the sake of completeness, let us now investigate the temporal evolution of the lightning activity in LCC(>20 ms)-producing
thunderstorms. Following the clustering method mentioned in Section 3.3, we cluster the lightning flashes reported by ENTLN
and taking place in the Iberian Peninsula in 2017 for 3 selected LCC(>20 ms)-producing thunderstorms reported by ISS-LIS.
Fig. 20 shows the cumulative number of flashes occurring in the selected cases 1400 minutes around the occurrence of a

LCC(>20 ms)-lightning flash. As in the case of fire-producing lightning discharges, *LCC(>20 ms)-lightning flashes tend to*
*occur at the moment when the lightning activity increase from the regime with the lowest lightning activity to the regimen with*
*the highest lightning activity*. We have found no relationships between the cumulative number of different type and polarity of
lightning discharges and the time of occurrence of LCC(>20 ms)-lightning.

### 3.5   Possible relationship between thunderstorms producing fire-igniting lightning and LCC lightning

Let us now compare the meteorological and cloud conditions of fire-igniting lightning and LCC(>20 ms)-lightning. According
to Fig. 18, there exists a sub set of LCC(>20 ms)-lightning flashes taking place in high-based clouds (CBH larger than 2000 m).
LCC(>20 ms)-lightning flashes are plotted as red dots in the first panel of Fig.15. As we discussed in Section 3.2, high-based
thunderclouds favor the occurrence of dry lightning and LIW. For this reason, in this section we analyze the meteorological
conditions of LCC(>20 ms)-lightning flashes taking place in high-based clouds with CBH values larger than 2000 m. The total

number of LCC(>20 ms)-lightning flashes in high-based thunderclouds is 392.

Fig. 21 shows the frequency distribution of some meteorological variables for typical lightning and LCC(>20 ms)-lightning
flashes in high-based thunderclouds. According to this figure, the median values of accumulated precipitation and RH at





850 hPa are lower for LCC(>20 ms)-lightning flashes in high-based thunderclouds than for the climatological median, with p-values lower than the threshold of 0.05. In addition, temperature at 2 m altitude and at 850 hPa are higher for LCC(>20 ms)-

lightning flashes in high-based thunderclouds than for the climatological median, again with p-values lower than the threshold of 0.05, indicating that differences are statistically significant. These are typical conditions of fire-igniting lightning flashes (see Section 3.2). We have not found statistically significant differences between the median value of CAPE for typical lightning and LCC(>20 ms)-lightning flashes (p-value greater than 0.05).

     Comparison of Figures 10, 11 and 17 shows that fire-igniting lightning flashes and LCC(>20 ms)-lightning flashes take

place in thunderstorms with weaker updraft than the climatological median, at least between 300 hPa and 450 hPa (where p-values lower than 0.05). Deep convective clouds are usually associated with the 440 hPa level (Rossow, 1996), included in the region where the updraft is weaker than the climatological median for LIW and LCC(>20 ms)-lightning. Several lightning parameterizations are based on the updraught mass flux at 440 hPa, where electrification occurs Grewe et al. (2001); Allen and Pickering (2002); Finney et al. (2014). Weaker updraft leads to a slower rime accretion rate (Takahashi, 1978). As suggested

by Bitzer (2017), weaker charging rate can lead to development of larger charge regions before lightning is initiated, producing a larger charge region available to be neutralized by a lightning flash. Large charge regions can favor the occurrence of LCC(>20 ms)-lightning (Bitzer, 2017) and, according to our results, could also favor the occurrence of LIW.

## 4   Discussion

To the best of our knowledge, this study is the first that investigates the shared meteorological characteristics of LIW and LCC-

lightning and the upper-tropospheric meteorological characteristics of LIW and LCC-lightning flashes in the Mediterranean. As extensively discussed in Section 1, past studies have reported the meteorological characteristics in the lower troposphere of LIW in the Mediterranean basin and the possible linkage between dry thunderstorms and fire-igniting flashes in the United States and in Australia (Rorig et al., 2007; Dowdy and Mills, 2012; Bates et al., 2017; Dowdy, 2020). According to our results, dry thunderstorms with high-based thunderclouds are also the main precursor of LIW in the Mediterranean basin. In accordance

with previous studies in Catalonia (e.g., Pineda and Rigo, 2017), our results also suggest that thunderstormstaking place with high temperatures at surface and at 850 hPa pressure level and horizontal wind velocities at surface tend to be related with LIW.

     Fuquay et al. (1967) proposed that LCC-lightning could be the main precursors of LIW. Bitzer (2017) reported the first climatology of LCC-flashes, without including the Mediterranean basin. According to Bitzer (2017), LCC-lightning tends to

occur in thunderstorms with weaker updraft than the climatological median. We have found that LIW and LCC(>20 ms)-lightning over the Mediterranean basin tend to occur in thunderstorms with weaker updrafts than the climatological median. In addition, we have found that a significant number of LCC(>20 ms)-lightning flashes take place in dry thunderstorms with high-based thunderclouds. According to our results, LIW and LCC(>20 ms)-lightning tend to occur at the moment when the lightning activity just starts to increase from the regime with the lowest lightning activity to the regimen with the highest

lightning activity. Our findings suggest a possible link between LIW and LCC(>20 ms)-lightning.





Some previous studies have developed statistical methods, machine learning approaches or parameterizations based on meteorological variables to predict LIW (e.g., Fuquay, 1979; Flannigan and Wotton, 1991; Krause et al., 2014; Coughlan et al., 2021). These methods typically use lightning parameterizations to predict the occurrence of lightning and then estimate the risk of LIW according to environmental factors. However, the employed lightning parameterizations do not distinguish between
regular/typical lightning and LCC-lightning flashes. The analysis of meteorological conditions of LIW and LCC(>20 ms)-lightning in the Mediterranean basin suggests that using the updraft between 300 hPa and 450 hPa could serve to improve LIW forecasting methods. The common features of thunderstorms producing LIW and LCC(>20 ms)-lightning flashes confirm that monitoring the occurrence of LCC(>20 ms)-lightning flashes and/or the characteristics of thunderstorms producing them could improve the forecast of fire ignition. Parameterization of LCC(>20 ms)-lightning flashes in atmospheric models based on these
features can serve to produce better estimate of the global fire emissions in a warmer climate.However, more data are needed to study the role of LCC(>20 ms)-lightning in LIW in other regions of the world.

## 5  Conclusions

The main objective of this work was to shed light into the following questions: What is the relationship between the occurrence of dry thunderstorms and LIW fires in the Mediterranean basin? What is the role of Long-Continuing-Current (LCC) lightning
flashes in the ignition?

We have investigated the meteorological and cloud conditions of fire-igniting lightning over the Iberian Peninsula and Greece between 2009 and 2017 and between 2017 and 2019, respectively. In addition, we have analyzed for the first time the relationship between LCC(>20 ms)-lightning determined from by ISS-LIS data over Europe with a continuous current lasting more than 20 ms and fire-igniting lightning. We have focused our analysis on LCC(>20 ms)-lightning flashes because the continuous
phase of the discharge can transfer a significant amount of energy to trees, producing ignition. We have developed the first LCC(>20 ms)-lightning map over the Mediterranean basin based on optical measurements of ISS-LIS between 2017 and 2020. However, more data are needed to produce a climatology.

We have searched the lightning candidate of each fire using a proximity index that combines the holdover and the distance between flashes and fires. The fire databases used in this study have been provided by national institutions from Spain, Portugal,
France and Greece, while the lightning data have been provided by WWLLN, ENTLN and ISS-LIS.. We have collected meteorological and cloud conditions for the lightning flashes included in this study from ERA5-reanalysis data set, from the Spanish State Meteorological Agency (AEMET) and from the CTH-product derived from Meteosat Second Generation (MSG) satellites measurements. Finally, we have compared the meteorological and cloud conditions of fire-producing lightning and LCC(>20 ms)-lightning producing thunderstorms with the climatological median. The main conclusions of this work are as
follows:

1. Fire-igniting flashes over the studied region tend to occur in dry thunderstorms with a low accumulated precipitation value and a low content of moisture. Fire-producing thunderstorms are characterized by high-based clouds and by weak updrafts between 300 hPa and 450 hPa pressure levels. High temperatures at surface and at 850 hPa pressure level and





horizontal wind velocities at surface favor the ignition and spreading of lightning-produced fires. We have obtained
a slightly higher probability of ignition for positive CG lightning flashes than for negative CG lightning flashes, in
agreement with Latham and Williams (2001).

2. Both fire-igniting flashes and LCC(>20 ms)-lightning flashes tend to occur during the brief transition phase when the
lightning activity increase from the regimen with the lowest lightning activity to the regimen with the highest lightning
activity.

3. The common meteorological and cloud conditions of thunderstorms producing LIW and LCC(>20 ms)-lightning flashes
are the low content of moisture and the weak updrafts between 300 hPa and 450 hPa pressure levels.

4. We have found that the temperature at 2 m altitude and at 850 hPa pressure level is higher than the climatological
value in high-based thunderclouds (CBH larger than 2000 m) producing LCC(>20 ms) flashes, coinciding with the
meteorological conditions of fire-producing thunderstorms. In total, about a third of all the analyzed LCC(>20 ms)-
lightning flashes occurs in high-based thunderclouds. This result suggests that parameterizing and/or monitoring the
occurrence of LCC(>20 ms)-lightning flashes could serve to improve fire forecasting models and LIW parameterizations.

Our analysis suggests that LCC-lightning occurrence over Europe can be parameterized in atmospheric models using mete-
orological variables as proxy. Such a parameterization can be used in future studies to improve the modeling of fire occurrence
and its atmospheric emissions in such models, where different atmospheric variables are used as proxies for the occurrence
of lightning [e.g., (Tost et al., 2007; Murray et al., 2012; Gordillo-Vázquez et al., 2019)]. The launch of the Meteosat Third
Generation (MTG) geostationary satellites of the EUropean organization for the exploitation of METeorological SATellites
(EUMETSAT) in 2022 will provide for the first time a continuous monitoring of the occurrence of lightning flashes and fires
in Europe and Africa through the instruments Lightning Imager (LI) and Flexible Combined Imager (FCI) from 2023, after
the commissioning phase (Stuhlmann et al., 2005). MTG-LI will also provide for the first time a climatology of LCC flashes
over Europe and Africa, enabling us to investigate the relationships between LIW and LCC flashes in both continents. New
flash and fire climatology provided by the MTG-LI and the MTG-FCI together with meteorological measurements will mean
a substantial advance in the study of the meteorological conditions of LIW in the Mediterranean basin.

*Data availability.* All data used in this paper are directly available after a request is made to authors F. J. P. I (FranciscoJavier.Perez-
Invernon@dlr.de) or H. H. (Heidi.Huntrieser@dlr.de). All the analyzed EUMETSAT CTH product are freely accessible through the EUMET-
SAT Earth Observation Portal (https://eoportal.eumetsat.int/userMgmt/protected/welcome.faces). The ERA5 meteorological data are freely
accesible through Copernicus Climate Change Service (C3S) (2017): ERA5: Fifth generation of ECMWF atmospheric reanalyses of the
global climate . Copernicus Climate Change Service Climate Data Store (CDS) (https://cds.climate.copernicus.eu/cdsapp). WWLLN light-
ning data are available from the University of Valencia. ENTLN lightning data are available from Earth Networks. ISS-LIS lightning data
are freely accessible through the NASA Global Hydrology Resource Center (https://ghrc.nsstc.nasa.gov/lightning/data/data{_}lis{_}iss).
Echotop radar measurements are available through the AEMET. Fire data over Spain are available through the Spanish Ministerio de Agri-



cultura, Pesca y Alimentación. Fire data over Portugal are freely available through ICNF Sistema de Gestao de Incendios Florestais (http://www2.icnf.pt/portal/florestas/dfci/inc). Fire data over France are freely available through Promethee database (https://www.promethee.com). Fire data over Greece are provided by the Hellenic Fire Brigade.

*Author contributions.* F.J.P.I.: Conceptualization, methodology, validation, formal analysis, investigation, data curation, writing—original draft. H.H.: Conceptualization, methodology, validation,, formal analysis, supervision, investigation, writing—review and editing. F.J.G.V.: Conceptualization, data curation, validation, writing—review andediting. N.P., J.M. and O.V.: Conceptualization, validation, writing—review and editing. S.S., J.N.G. V.R. and N.K.: data curation.

*Competing interests.* Authors declare no competing interests.

*Acknowledgements.* The authors would like to thank NASA for providing LIS-ISS lightning data, Earth Networks for providing ENTLN lightning data, WWLLN for providing WWLLN lightning data, Spanish State Meteorological Agency (AEMET) for providing echotop radar measurements, EUMETSAT for providing CTH product data, ECMWF for providing us the data of ERA5 forecasting models, the Spanish Ministerio de agricultura, Pesca y Alimentación for providing fire data over Spain, the Instituto da Conservaçao da Naturaleza e das florestas for providing fire data over Portugal, the Prométhée database for providing fire data over Mediterranean France and the Hellenic Fire Brigade for providing fire data over Greece. Authors would also like to thanks Luca Bugliaro Goggia (Deutsches Zentrum für Luft- und Raumfahrt, DLR) for his support in the processing of METEOSAT data and Matthias Nützel (Deutsches Zentrum für Luft- und Raumfahrt, DLR) for providing valuable comments on this manuscript. FJPI acknowledges the sponsorship provided by the Federal Ministry for Education and Research of Germany through the Alexander von Humboldt Foundation. Additionally, this work was supported by the Spanish Ministry of Science and Innovation, under projects ESP2017-86236-C4-4-R, PID2019-109269RB-C43 and FEDER program. FJGV and SS acknowledge financial support from the State Agency for Research of the Spanish MCIU through the 'Center of Excellence Severo Ochoa' award for the Instituto de Astrofísica de Andaluca (SEV-2017-0709). The contribution of the Universitat Politecnica de Catalunya was supported by research grants from the Spanish Ministry of Economy and the European Regional Development Fund (FEDER): ESP2017-86263-C4-2-R and PID2019-109269RB-C42.



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





# Iberian Peninsula



**Figure 5.** Frequency distribution of the Convective Available Potential Energy (CAPE), the hourly accumulated precipitation, the horizontal wind at surface, the relative humidity at 850 hPa level and the air temperature at 2 m and at 850 hPa level for the CG lightning and the fire-igniting lightning climatologies in the Iberian Peninsula between 2009 and 2015.





**Figure 6.** Frequency distribution of the Convective Available Potential Energy (CAPE), the hourly accumulated precipitation, the horizontal wind at surface, the relative humidity at 850 hPa level and the air temperature at 2 m and at 850 hPa levels for the CG lightning and the fire-igniting lightning climatologies in Greece between 2017 and 2019.





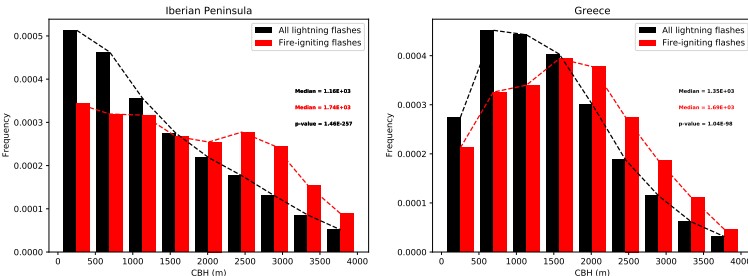

**Figure 7.** Frequency distribution of the CBH value for the CG lightning and the fire-igniting lightning climatologies in (left) the Iberian Peninsula between 2009 and 2015 and (right) in Greece between 2017 and 2019.

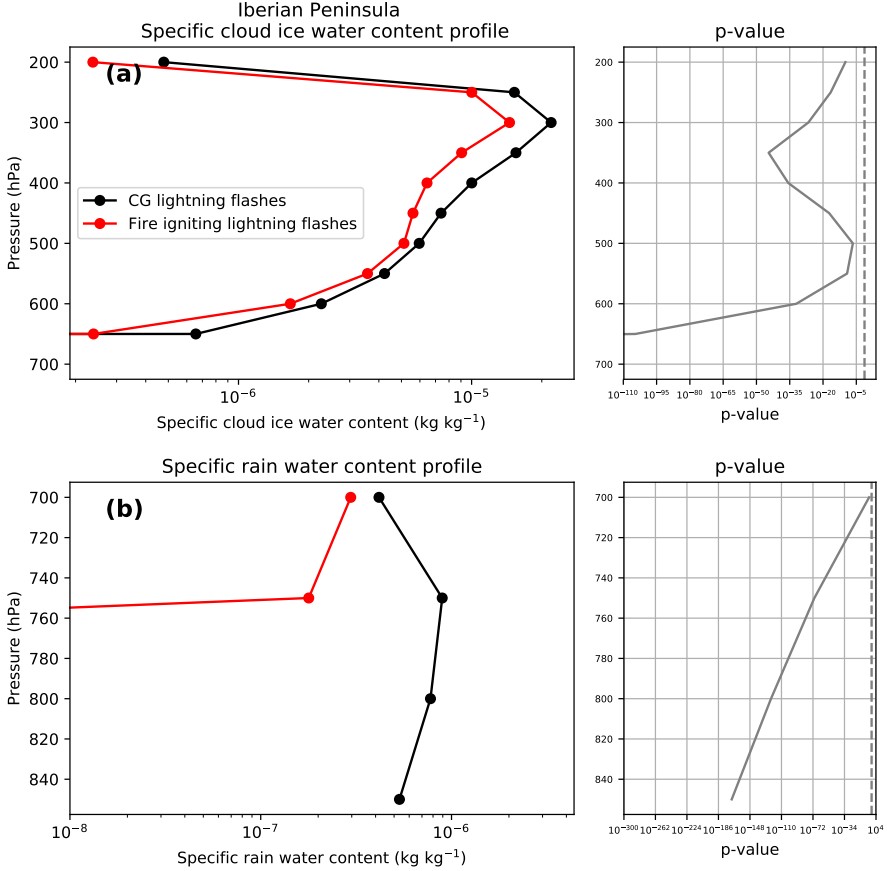

**Figure 8.** First column shows the vertical profiles of the specific cloud ice water content (a) and the specific rain water content (b) for the CG lightning and the fire-igniting lightning climatologies during May and September in the Iberian Peninsula between 2009 and 2015. Second column shows the $p-value$ (solid line) for each vertical level representing the probability of equal median between both distributions and a mark showing the limit at 0.05 (dashed line).



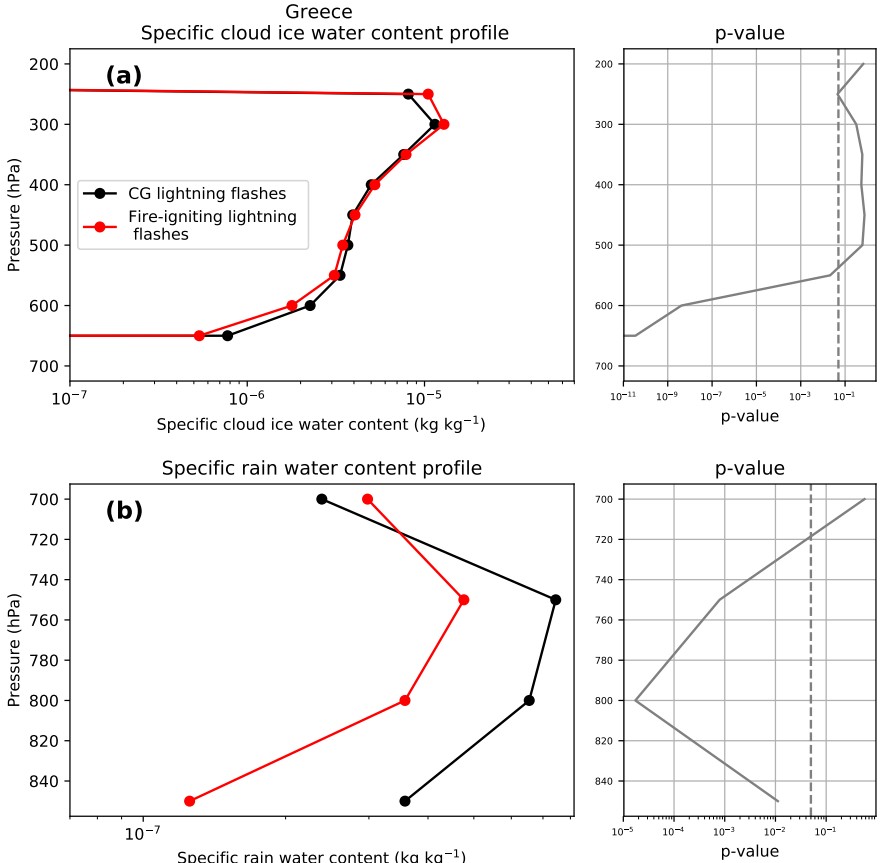

**Figure 9.** First column shows the vertical profiles of the specific cloud ice water content (a) and the specific rain water content (b) for the CG lightning and the fire-igniting lightning climatologies during May and September in Greece between 2017 and 2019. Second column shows the $p-value$ (solid line) for each vertical level representing the probability of equal median between both distributions and a mark showing the limit at 0.05 (dashed line).




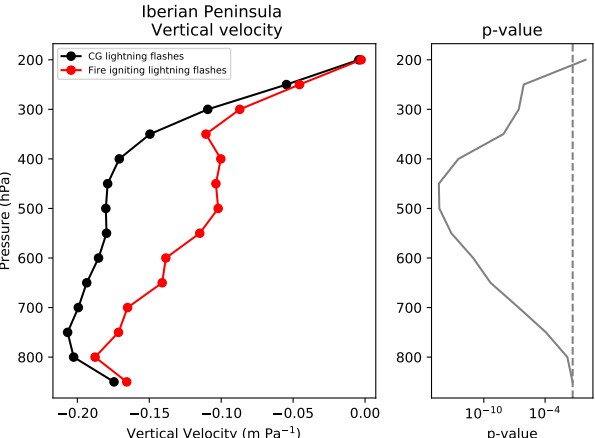

**Figure 10.** Median vertical velocity profile for the CG lightning and the fire-igniting lightning climatologies in the Iberian Peninsula between May and September for the period between 2009 and 2015 (left panel). $p-value$ (solid line) representing the probability of equal median between both distributions and mark showing the limit at 0.05 (dashed line) (right panel).

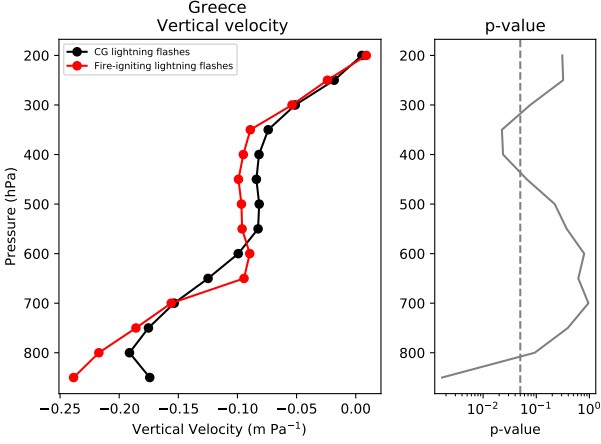

**Figure 11.** Median vertical velocity profile for the CG lightning and the fire-igniting lightning climatologies in Greece between May and September for the period between 2017 and 2019 (left panel). $p-value$ (solid line) representing the probability of equal median between both distributions and mark showing the limit at 0.05 (dashed line) (right panel).



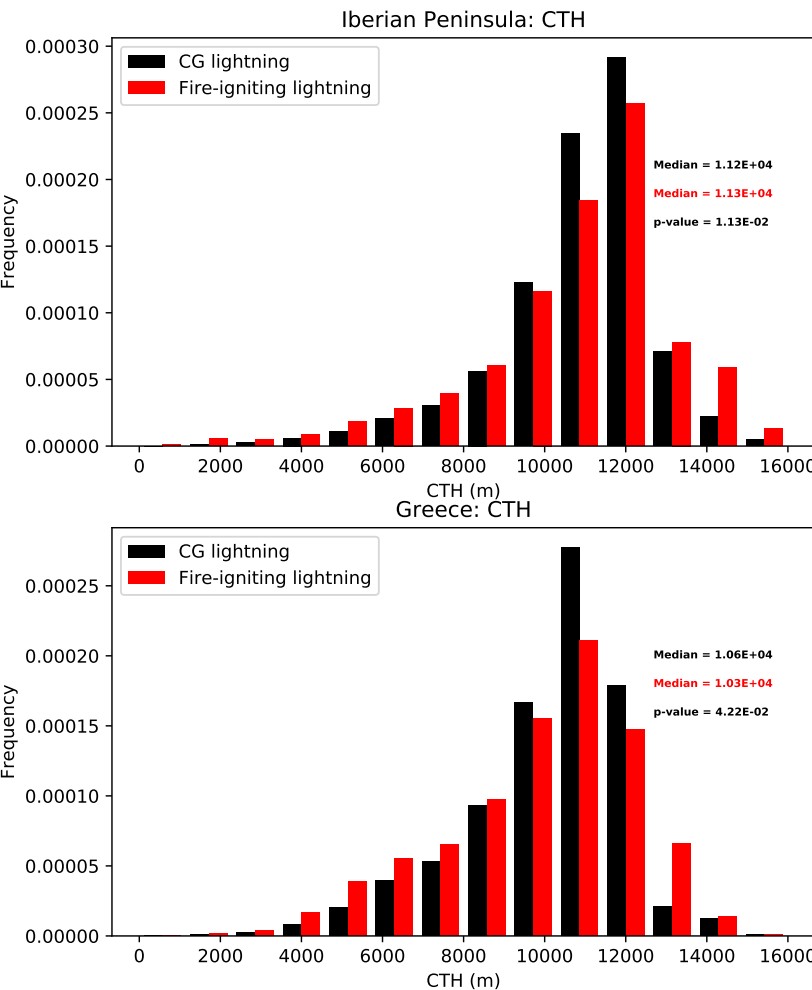

**Figure 12.** Frequency distribution of the CTH values reported by EUMETSAT based on measurements of the MSG satellites for fire-igniting lightning and all CG lightning over coniferous and mixed forests in the Iberian Peninsula (2012-2015) and Greece (2017-2019).



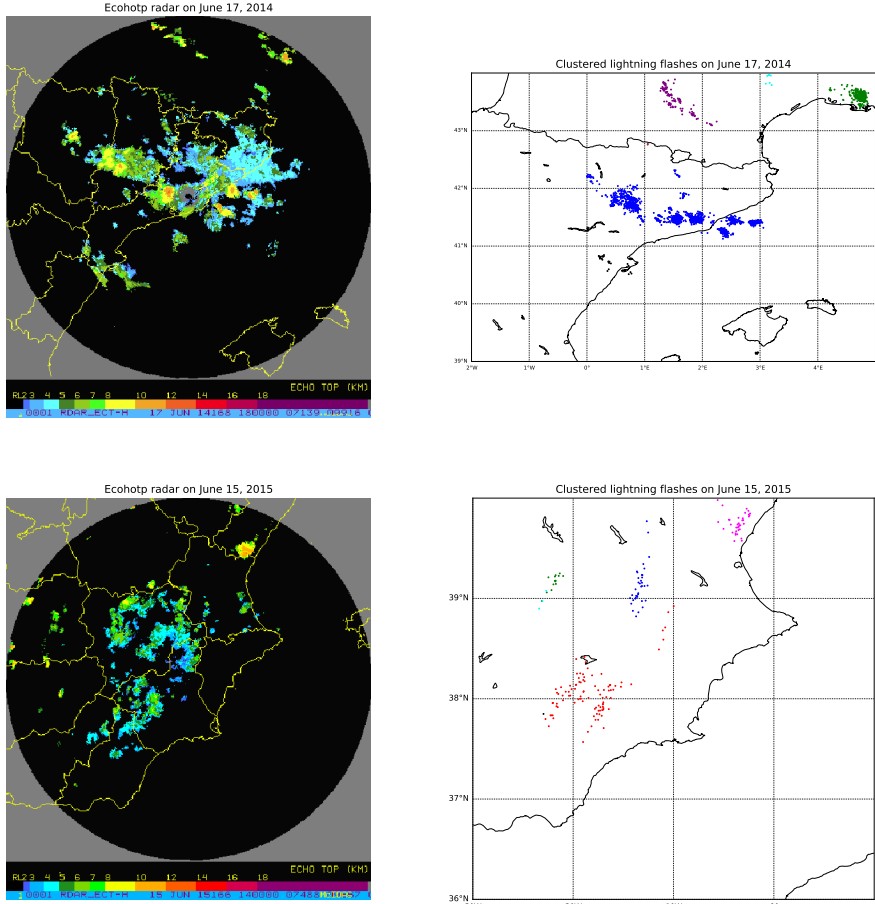

**Figure 13.** Echo top radar images provided by AEMET showing the maximum height in km of precipitable particle echoes at 12 dB and 15 minutes around one fire-ignited lightning on June 17, 2014 at 18:00 UTC (upper left panel) and on June 15, 2015 at 14:15 UTC (lower left panel). Lightning flashes reported by ENTLN on June 15 2014 between 17:45 and 18:15 UTC (upper right panel) and on June 15, 2015 between 14:00 and 14:30 UTC (lower right panel). Different colors indicate different lightning clusters or thunderstorms using ST-DBSCAN with the spatial and temporal parameters 0.12° and 15 minutes, respectively.



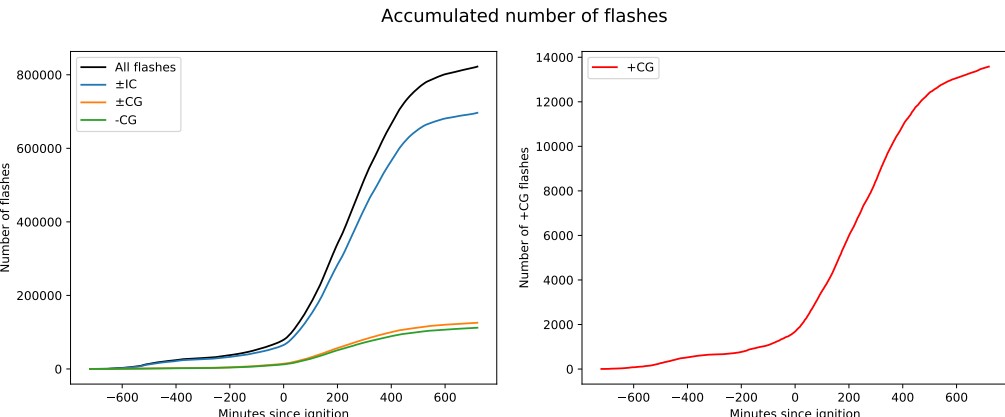

**Figure 14.** Cumulative number of flashes occurring in 77 fire-igniting thunderstorms over the Iberian Peninsula 1400 minutes around the ignition of a lightning-ignited fire. We plot all flashes, all IC flashes (±IC), all CG flashes (±CG) and -CG flashes in the right panel. We plot +CG flashes in the left panel.

**Figure 15.** LCC(>20 ms)-lightning flash density (left column), total lightning flash density (right column) and ratio between them (lower center panel) from ISS-LIS lightning measurements between March 2017 and September 2020. First row shows maps over Europe with $1° × 1°$ , while second and third right rows show zooms over the regions of interest (only flashes over land) $0.5° × 0.5°$. Red dots in the left row represent the LCC(>20 ms)-lightning flashes taking place over land and with CBH larger than 2000 m. For the sake of clarity, in the map of Europe (lower center panel) the ratio in one cell is set to zero if there are less than 10 lightning flashes and less than 2 LCC(>20 ms)-lightning flashes.



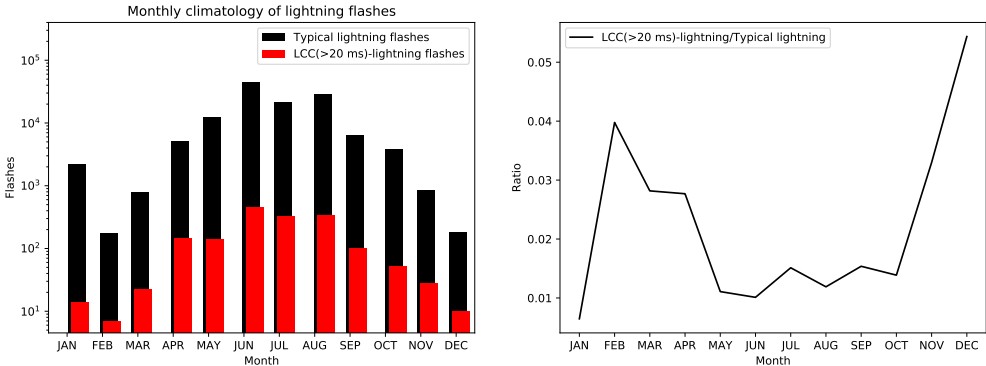

**Figure 16.** Left panel shows the monthly occurrence of typical lightning and LCC(>20 ms)-lightning flashes reported by ISS-LIS over land in Europe between March 2017 and September 2020. Right panel shows the monthly averaged ratio of LCC(>20 ms)-lightning flashes to all flashes.

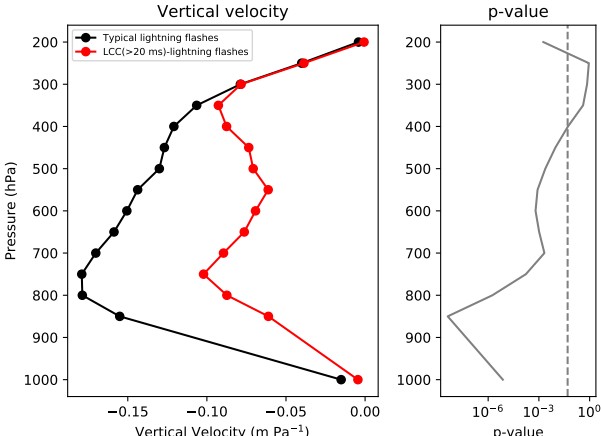

**Figure 17.** Median vertical velocity profile of typical lightning flashes and LCC(>20 ms)-lightning flashes reported by ISS-LIS over Europe between May and September in 2017-2020 (left panel). $p-value$ (solid line) representing the probability of equal median between both distributions and mark showing the limit at 0.05 (dashed line) (right panel).





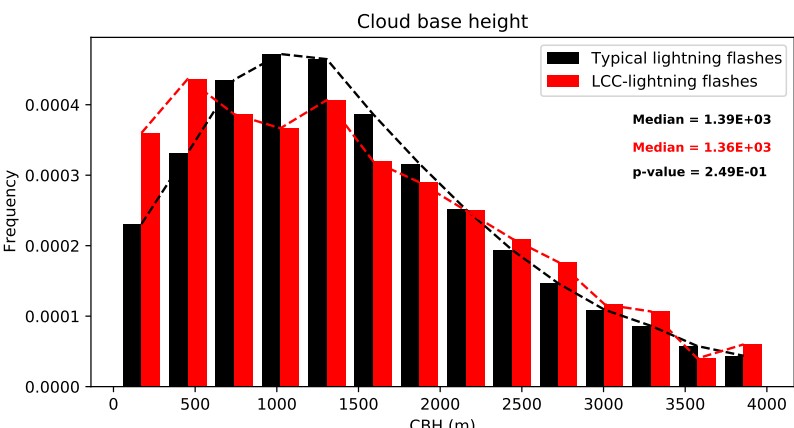

**Figure 18.** Frequency distribution of the CBH value for typical lightning flashes and for LCC(>20 ms)-lightning flashes reported by ISS-LIS over Europe between May and September in 2017-2020.

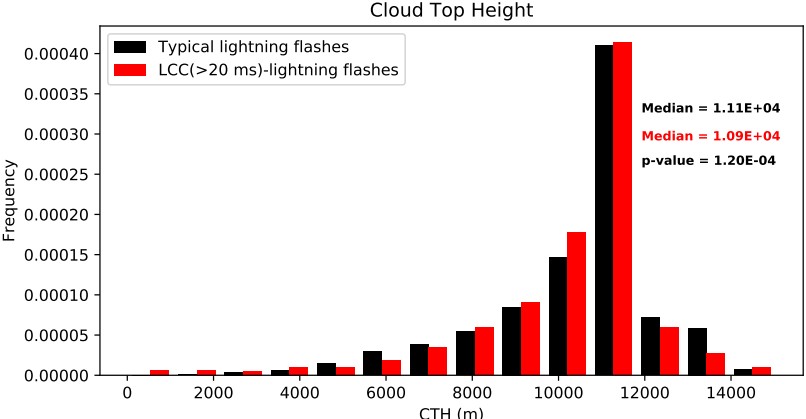

**Figure 19.** Frequency distribution of the CTH values reported by EUMETSAT based on measurements of the MSG satellites for typical lightning flashes and LCC(>20 ms)-lightning flashes reported by ISS-LIS over Europe between March and September in 2017-2019.



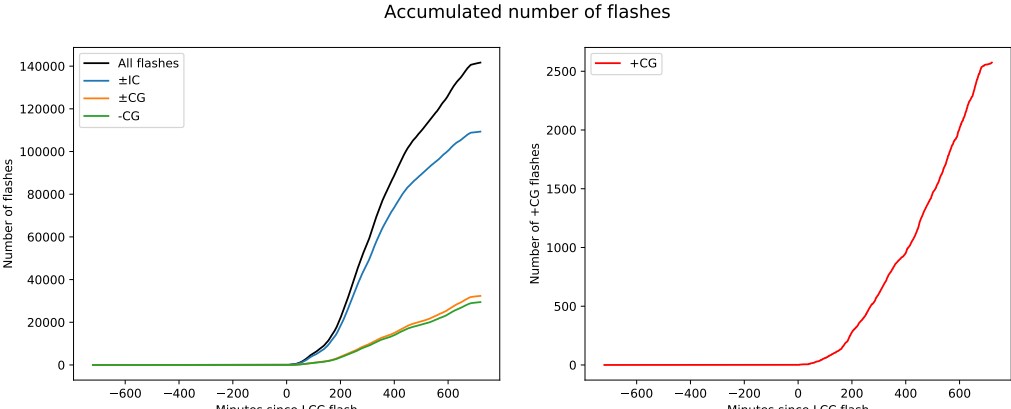

**Figure 20.** Cumulative number of flashes occurring in 3 LCC(>20 ms)-producing thunderstorms 1400 minutes around the occurrence of the LCC lightning flash. We plot all flashes, all IC flashes (±IC), all CG flashes (±CG) and -CG flashes in the right panel. We plot +CG flashes in the left panel.



**Figure 21.** Frequency distribution of the Convective Available Potential Energy (CAPE), the hourly accumulated precipitation, the relative humidity at 850 hPa level and the air temperature at 2 m and at 850 hPa levels for typical lightning and LCC(>20 ms)-lightning flashes in high-based thunderclouds (CBH > 2000 m) reported by ISS-LIS over land in Europe between May and September 2017-2020.