# Peer review of "Lightning-ignited wildfires and long-continuing-current lightning in the Mediterranean Basin: Preferential meteorological conditions"

_Atmospheric Chemistry and Physics, 2021_

## Referee Comment (RC1)

May 2nd 2021

ACP

Dear Sir, Madam

Review of: Lightning-ignited wildfires and long-continuing-current lightning in the Mediterranean Basin: Preferential meteorological conditions by Francisco J. Pérez-Invernón et al. (ACP-2021-125)

I have now completed the review of the paper by Pérez-Invernón et al. submitted for publication in ACP. This is a good, timely and comprehensive paper, that addresses an interdisciplinary topic with observational and analytical tools. The authors combine various data types from different platforms and conduct a thorough analysis aiming to distinguish and identify the types of meteorological conditions prevalent in Greece and the Iberian Peninsula which are conducive to producing lightning-ignited forest fires. The topic is highly relevant to the readership of ACP.

The graphs and tables are adequate, and the paper is well-organized, the language is fluent and clear (some typos here and there) and the overall quality of presentation is very good.

I have several that relate to the analysis and the conclusions, which I present for the authors to respond. They may require a minor revision of the manuscript before being accepted for publication.

Major Comments

1. Section 3.4.1 describes the methodology of obtaining LCC(>20 ms) from LIS data. However, it is entirely possible that these strokes are IC flashes and unrelated to LIW. Have the authors compared those events with ENTLN or WWLLN lightning data for specific storms that ignited wildfires? Were they really CG strokes? It would be more convincing if indeed Long Continuing Current strokes are also detected by ground systems and a correlation between brightness duration and actual peak-current or energy is obtained. [With this in mind, could it be that LCC(>20 ms) strokes are superbolts? (Holzworeth et a., 2019)]. Are the authors able to define the multiplicity of flashes with LCC(>20 ms)?

2. In trying to reconcile the dynamical and microphysical structure of thunderstorms that ignite fires compared with those that do not, there seems to be a contradiction (or at least, inconsistency) between the depth of storms as defined by their average CTH reported by satellites (Figure 12) and the fact that on average they exhibit slower updrafts (Figures 10) or faster (Figure 11). This fact also seems at odds with the statement (line 338) that the instability is higher for clouds that produce fire-igniting strikes compared with those that do not. This is also mentioned in section 3.2.1 with regards to CAPE values (line 360) where fire igniting lightning in the Iberian Peninsula have lower CAPE values compared with the climatological media.

It is a well-known fact that supersaturation closely depends on the vertical velocity (see Rogers and Yau, 3$^{rd}$ edition 1989, chap. 6) and so one would expect that slower updrafts will result in less activated CCN, less droplets and fewer ice crystals, all leading to a reduced efficiency in charging. Can the authors elucidate this mismatch between dynamics and microphysics?

3. Lines 455-468: The geographical distribution of LCC (>20 ms) with Cloud Base Height (CBH > 2km) as presented in Figure 15 shows that they are produced mainly over land, and not as stated in the text over the ocean and in coastal areas, even when the total lightning is over land (line 465). This is in contrast with the cited Holzworth et al (JGR, 2019) paper and with Fullekrug et al. (Ann. Geophys., 20, 133–137, 2002) that showed intense lightning (or super-bolts) to be occurring over oceans and near coasts. At least this is what this reviewer sees in the upper panel of Figure 15. Am I missing something here? If the most intense lightning indeed occurs in coastal areas and above sea water, how can they be the ones that ignite forest fires? This seemingly contradictory results is actually discussed in lines 469-474. Further explanation is needed.

4. The distinction between storms that produce lightning with LCC (>20 ms) and those that produce only LCC (>10 ms) and "normal" ones is not entirely clear to me. Let us suppose that there was just 1 flash with a long continuing current – does this qualify the storm to be included in the statistics? Or is there a threshold of some number of such flashes? After all, lightning discharge processes are (almost) entirely random and it can well be that a storm has all the "ingredients" needed to produce LLC (>20 ms) and still does not. This randomness is partially manifested in the seasonal ratio as described by Figure 16, which is higher in winter. Nevertheless, winter thunderstorms produce fewer flashes and are generally less deep and so (in line with comments #1) may not be ideal for generating such flashes.

5. The weakest part of the paper is the concept of the "transition phase" discussed in section 3.4.3 (and also in lines 602-604). The definition is somewhat unclear, and is unrelated to the typical microphysics and dynamical evolution of thunderstorms. If there is a clear change from a low-flash rate to a high-flash rate regime (or vice-versa) prior to the occurrence of LCC(>20 ms) strokes then we should see specific quantitative values describing these phases of the storm. For example, Emersic et al. (MWR, 2011) defined 3 distinct periods of lightning activity, and related them to the charge structure (see their Figures 4 and 5). Lang et al. (BAMS, 2004) showed how the flash-rate evolves as a function of time while differentiating between IC and CG strokes along the storm's life cycle. It is unclear how LCC(>20 ms) strokes are distributed as a function of time and if (and how) they are related to cloud microphysics. Either give more information or delete this section.

6. In lines 469-479 the authors discuss the comparison between LIW maps and LCC(>20 ms) maps. There are several places where we see fires, but no strong lightning. What can be the interpretation of these fire events? Is there a possibility that those fires had been ignited by "regular" strokes, or those with shorter CC? It seems that the selection of 20ms threshold is arbitrary, and

actually there may be episodes that even shorter strokes can ignite fires (for example if the forest was dry or already deteriorated).

7. The discussion about forecasting the potential for LIW (lines 565-575) may benefit from including the concept of the Lightning Potential Index (LPI; Yair et al., JGR 2010). This parameter was later developed into the Dynamic Lightning Index by Lynn et al. (WAF, 2012). Perhaps simulating LIW events and "calibrating" the LPI values against the occurrence of LCC(>20 ms) will improve forecast capabilities in operational models.

---

## Author Comment (AC1)

**Rebuttal**

We would like to thank the reviewers for their thoughtful comments and efforts towards improving our manuscript. We address comments specific to reviewer 1 below (blue letters).

**Reviewer 1**

I have now completed the review of the paper by Pérez-Invernón et al. submitted for publication in ACP. This is a good, timely and comprehensive paper, that addresses an interdisciplinary topic with observational and analytical tools. The authors combine various data types from different platforms and conduct a thorough analysis aiming to distinguish and identify the types of meteorological conditions prevalent in Greece and the Iberian Peninsula which are conducive to producing lightning-ignited forest fires. The topic is highly relevant to the readership of ACP.

The graphs and tables are adequate, and the paper is well-organized, the language is fluent and clear (some typos here and there) and the overall quality of presentation is very good.

We thank the referee for these encouraging comments.

I have several comments that relate to the analysis and the conclusions, which I present for the authors to respond. They may require a minor revision of the manuscript before being accepted for publication.

Major Comments

1. Section 3.4.1 describes the methodology of obtaining LCC(>20 ms) from LIS data. However, it is entirely possible that these strokes are IC flashes and unrelated to LIW. Have the authors compared those events with ENTLN or WWLLN lightning data for specific storms that ignited wildfires? Were they really CG strokes? It would be more convincing if indeed Long Continuing Current strokes are also detected by ground systems and a correlation between brightness duration and actual peak-current or energy is obtained. [With this in mind, could it be that LCC(>20 ms) strokes are superbolts? (Holzworeth et al., (2019)]. Are the authors able to define the multiplicity of flashes with LCC(>20 ms)?

   We agree with the reviewer that some of the detected LCC-lightning could indeed be IC instead of CG lightning. Bitzer (JGR, 2017) compared the LCC-lightning detected by TRMM-LIS over United States with lightning strokes reported by NLDN and found:
   "25–40% of flashes identified by LIS to have continuing current have only an intracloud pulse detected by the National Lightning Detection Network (NLDN), with no cloud-to-ground strokes detected."

   Before comparing the LCC-lightning detected by ISS-LIS in the Iberian Peninsula and Greece with lightning reported by ENTLN, we have to take the following two points into consideration:

   > 1) The quasi-electrostatic field produced by the continuing phase of LCC-lightning flashes decays strongly with the distance ($1/r^3$), while the impulsive phase decays as ($1/r^2$). Therefore, Extreme Low Frequency (ELF) sensors are needed to report the

continuing phase of the discharge if the flash is far from the sensors. It is expected that VLF ground-based lightning detection networks (such as ENTLN or WWLLN) do not report the continuing phase of the discharge. As a consequence, we can expect that a part of all the CG LCC-flashes are missed if they are not preceded by an impulsive phase.

2) Pineda et al. (2014) reported optical and radio signals of 94 negative strokes and 10 positive strokes in Catalonia (part of the regions studied in this work). They found that all negative LCC-lightning had a peak current below ~20 kA. Such low impulsive lightning flashes can be missed by Lightning Location Systems, such as ENTLN and WWLLN, tend to miss-classified small positive peak current strokes [Cummins and Murphy, 2009].

We have now compared the LCC(>20 ms) lightning flashes with CBH>2 km taking place in the Iberian Peninsula and Greece with ENTLN data. Among 39 LCC(>20 ms) lightning flashes reported by ISS-LIS, ENTLN detected only 12 lightning flashes. ENTLN does not provide information about the most of the LCC-lightning detected by ISS-LIS. Therefore, we conclude that ENTLN is not adequate to detect the continuing phase of LCC(>20 ms) flashes in the studied regions because it missed more than a half of all the LCC-lightning. Of the 12 flashes detected by ENTLN, 3 were CG and 9 were IC.

As reported by Pineda et al. (2014), we expect that all the negative LCC-lightning taking place in the area have a peak current below 20 kA. The detection efficiency of VLF sensors is lower for low impulsive lightning flashes. Therefore, ENTLN can have missed an important part of all the -CG LCC-lightning flashes.

We have also attempted to directly correlate LIW in Greece (2017-2019) and over the Iberian Peninsula (2017) with LCC-flashes reported by ISS-LIS. However, we did not find any case in which ISS-LIS was flying over Greece or the Iberian Peninsula at the moment of ignitions. **We have included this reasoning in the manuscript.**

Regarding the possibility of LIW being superbolts, we have analyzed the stroke energies in the VLF band between 5 and 18 kHz for WWLLN data in the Iberian Peninsula. We assume that strokes with energy above $10^6$ J are superbolts. We have found 95 superbolts over coniferous and mixed forest and between May and September over the Iberian Peninsula (among 207116 strokes) and 3 superbolts that are also fire-igniting lightning-candidates (among 1697 fire-igniting lightning-candidates). Therefore, 0.18% of fire-igniting lightning-candidates are superbolts, while 0.05% of all strokes in coniferous and mixed forests are superbolts. We can then conclude that superbolts incidence is higher for fire-igniting lightning than for typical lightning, although superbolts do not represent a significant part of fire-igniting lightning. However, as we have previously discussed and as reported by Pineda et al. (2014), we expect that a significant number of LCC flashes can be not reported if they are not followed by an impulsive phase, especially negative LCC flashes [Pineda et al. (2014)]. Therefore, the percentage of superbolts-igniting fires with respect to all fire-igniting lightning flashes could be larger than 0.18%. **We have included this finding in the**

**manuscript.** We do not have access to the total radiated energy by ENTLN flashes, so we only study superbolts reported by WWLLN.

Unfortunately, we do not have access to the energies of strokes between 2017 and 2020. Therefore, we cannot estimate the stroke energies of LCC(>20 ms) flashes in order to compare it with the reported superbolts-igniting fires. However, Holzworeth et al, (2019) suggested that superbolts tend to have a larger than averaged peak current in ENTLN data. We analyze the peak current of the 12 lightning flashes reported by ENTLN that coincide with  LCC(>20 ms) lightning flashes with CBH>2 km. The absolute mean peak current of these flashes is 8 kA, while the absolute mean peak current of all ENTLN flashes in the Iberian Peninsula between 2014 and 2015 is 14 kA. The peak current of the 3 CG flashes reported by ENTLN that coincide with  LCC(>20 ms) lightning flashes with CBH>2 km are -19 kA, -10 kA and -15 kA, while the absolute mean peak current of all CG flashes reported by ENTLN in the Iberian Peninsula between 2014 and 2015 is 25 kA. Therefore, we do not see any clear relationship between LCC(>20 ms) ligthning and superbolts. However, we agree with the reviewer that a more extensive analysis of the energy of LCC(>20 ms) lightning flashes using WWLLN data is interesting and could provide some insights into the global occurrence rate of LCC(>20 ms) lightning and their possible relationships with LIW.

2. In trying to reconcile the dynamical and microphysical structure of thunderstorms that ignite fires compared with those that do not, there seems to be a contradiction (or at least, inconsistency) between the depth of storms as defined by their average CTH reported by satellites (Figure 12) and the fact that on average they exhibit slower updrafts (Figures 10) or faster (Figure 11). This fact also seems at odds with the statement (line 338) that the instability is higher for clouds that produce fire-igniting strikes compared with those that do not. This is also mentioned in section 3.2.1 with regards to CAPE values (line 360) where fire igniting lightning in the Iberian Peninsula have lower CAPE values compared with the climatological media.

We agree with the reviewer that there seems to be a contradiction between the variables that we have used to discuss the instability of the atmosphere for LIW and typical lightning (CAPE, temperature vertical profile and CTH). Let us discuss about these contradictions.

Firstly, it is important to point out that we have found an error in the analysis of CAPE, CBH and CTH. We were wrongly removing some values from the sample of CG/typical lightning flashes. The median values of these variables have slightly changed without affecting the conclusions.

Let us now focus on the comparison between the instability as suggested by the vertical velocity, CAPE and the differences in temperature in the Iberian Peninsula. We have changed the altitudes at which we calculate the median temperature to discuss about atmospheric instability (line 338). Instead of using the pressure level 850 hPa, we have now used the level 700 hPa. We propose this modification because the pressure level 850 hPa can be too close to the surface for LIW over mountains and do not represent the state of the medium atmosphere. Now the manuscript reads:

"We have found that the median temperature differences between the surface (2~m altitude above ground) and 700~hPa pressure level for CG lightning in the Iberian Peninsula and Greece are, respectively, 16.8~K and 19.0~K. In the case of fire-igniting lightning, the median temperature differences are 18.8~K and 19.9~K, respectively. The median temperature differences between 700~hPa and 450~hPa pressure levels for CG lightning in the Iberian Peninsula and Greece are 22.4~K and 21.8~K, respectively, while in the case of fire-igniting flashes the median temperature differences are, respectively, 23.1~K and 21.8~K."

The received differences between 700 hPa and 450 hPa are only slightly higher for LIW over the Iberian Peninsula than for the climatological median (22.4 K vs 23.1 K). The stability of the atmosphere (CAPE and the vertical velocity) does not only depend on the temperature profile, but also on humidity. We have calculated the median specific humidity at 450 hPa (from ERA5 reanalysis, hourly data) for fire-producing lightning and CG lightning in the Iberian Peninsula, obtaining respectively 7.7e-4 kg kg$^{-1}$ and 8.65e-4 kg kg$^{-1}$ (p-value<0.05). Given that dry air is more dense than moist air, we suggest that the updraft of thunderstorms producing LIW over the Iberian Peninsula is lower than the climatological median because they are composed by air that is drier than the climatological median.

Finally, Fig. 12 suggests that the median value of CTH for LIW and for typical CG are similar (in Greece) or slightly higher for LIW than for typical lightning (in the Iberian Peninsula). As the reviewer points out, there seems to be a disagreement between the CTH and the updrafts in LIW. It is important to highlight that CTH data is reported every 15 min and with a horizontal resolution of 4 km by EUMETSAT, while ERA5 data is 1-hourly average with a resolution of 0.25x0.25 degrees. The cloud top height at a given point can significantly change in 1 h (Emersic et al. (MWR, 2011), Fig. 8) and within a 0.25x0.25 degrees cell. Therefore, LIW (or LCC(>20 ms)) can occur in thunderstorms with weaker convection than the averaged at the spatio-temporal scale of ERA5 but at the time and position when/where CTH is high.

**We have included this discussion in the manuscript.**

3. It is a well-known fact that supersaturation closely depends on the vertical velocity (see Rogers and Yau, 3$^{rd}$ edition 1989, chap. 6) and so one would expect that slower updrafts will result in less activated CCN, less droplets and fewer ice crystals, all leading to a reduced efficiency in charging. Can the authors elucidate this mismatch between dynamics and microphysics?

Bitzer, JGR (2017) provided a possible explanation for this possible mismatch between dynamics and microphysics:

"Oceanic and winter lightning are less frequent [e.g.,Cecil et al., 2012] but are more likely to contain continuing current. These storms typically have weaker updrafts, which suggest the likelihood that storms that have this characteristic are more likely to produce flashes with continuing current. The weaker updrafts suggest that charging rates are smaller, allowing for larger charge regions to develop before lightning is initiated. This in turn provides a larger charge region available to be neutralized by a flash. More work remains to

be done for a more definitive conclusion, however. A target study utilizing electric field change data, optical measurements, and radar could provide support for this hypothesis."

As we discussed in point 2, we are not able to determine if thunderstorms producing LIW and LCC(>20 ms) have weaker updrafts because of the spatio-temporal resolution of ERA5. However, according to our results, LIW and LCC(>20 ms) tends to occur when the updraft is weak. We think that the explanation of Bitzer, JGR (2017) is also valid to explain the apparent mismatch mentioned by the reviewer.

4. Lines 455-468: The geographical distribution of LCC (>20 ms) with Cloud Base Height (CBH > 2km) as presented in Figure 15 shows that they are produced mainly over land, and not as stated in the text over the ocean and in coastal areas, even when the total lightning is over land (line 465). This is in contrast with the cited Holzworth et al (JGR, 2019) paper and with Fullekrug et al. (Geophys., 20, 133–137, 2002) that showed intense lightning (or super-bolts) to be occurring over oceans and near coasts. At least this is what this reviewer sees in the upper panel of Figure 15. Am I missing something here? If the most intense lightning indeed occurs in coastal areas and above sea water, how can they be the ones that ignite forest fires? This seemingly contradictory results is actually discussed in lines 469-474. Further explanation is needed.

In Fig. 15 we only include LCC(>20 ms) flashes with CBH>2 km **over land**. This is why the reviewer cannot see any LCC(>20 ms) with CBH>2 km over ocean. We have not plotted LCC(>20 ms) with CBH>2 km over ocean because the main focus of splitting the sample according to CBH is to find relationships between those flashes and LIW. However, as the first row of Fig. 15 shows, there is a significant number of LCC(>20 ms) flashes over ocean.

5. The distinction between storms that produce lightning with LCC (>20 ms) and those that produce only LCC (>10 ms) and "normal" ones is not entirely clear to me. Let us suppose that there was just 1 flash with a long continuing current – does this qualify the storm to be included in the statistics? Or is there a threshold of some number of such flashes? After all, lightning discharge processes are (almost) entirely random and it can well be that a storm has all the "ingredients" needed to produce LLC (>20 ms) and still does not. This randomness is partially manifested in the seasonal ratio as described by Figure 16, which is higher in winter. Nevertheless, winter thunderstorms produce fewer flashes and are generally less deep and so (in line with comments #1) may not be ideal for generating such flashes.

We include a thunderstorm in the statistics as a LCC(>10 or 20 ms) producing thunderstorm if there is at least one LCC(>10 or 20 ms) flash reported **during ISS-LIS overpass** (the overpass from each point lasts about 90 seconds). On the contrary, thunderstorms without reported  LCC(>10 or 20 ms) flashes during ISS-LIS overpass are included in the statistic as "normal" thunderstorms.

As the reviewer points out, it is possible that a thunderstorm has all the ingredients needed to produce a LCC flash but still does not. It is also possible that a thunderstorm produces LCC flashes after/before ISS-LIS overpass and is misidentified as "normal" thunderstorm. However, at a global scale only ~1.71% of all lightning flashes are LCC (>20 ms) [Bitzer (JGR, 2017)], a number that would be even smaller over land (see Bitzer (JGR, 2017) Fig. 6). Therefore, we think that the effect of misidentifying LCC(>20 ms) producing thunderstorms as "normal" thunderstorms is not significant, as the group of "normal"

thunderstorms would then be dominated by the meteorological conditions of "normal" thunderstorms. At the same time, we think it is extremely unlikely that the fast-traveling ISS-LIS catches LCC(>20 ms) flashes from a thunderstorm producing only a few total number of LCC(>20 ms) flashes.

Bitzer, (JGR, 2017) also reports higher proportion of LCC flashes in oceanic and winter storms as in, respectively, oceanic and summer thunderstorms. This finding motivated us to hypothesize that thunderstorms producing LCC flashes could have different meteorological conditions than "normal" thunderstorms.

6.  The weakest part of the paper is the concept of the "transition phase" discussed in section 3.4.3 (and also in lines 602-604). The definition is somewhat unclear, and is unrelated to the typical microphysics and dynamical evolution of thunderstorms. If there is a clear change from a low-flash rate to a high-flash rate regime (or vice-versa) prior to the occurrence of LCC(>20 ms) strokes then we should see specific quantitative values describing these phases of the storm. For example, Emersic et al. (MWR, 2011) defined 3 distinct periods of lightning activity, and related them to the charge structure (see their Figures 4 and 5). Lang et al. (BAMS, 2004) showed how the flash-rate evolves as a function of time while differentiating between IC and CG strokes along the storm's life cycle. It is unclear how LCC(>20 ms) strokes are distributed as a function of time and if (and how) they are related to cloud microphysics. Either give more information or delete this section.

    As Emersic et al. (MWR, 2011) suggests, differentiation between successive thunderstorms entails important difficulties. We have differentiated thunderstorms according to the lightning clustering method explained in the draft without meteorological considerations. As a consequence, we have obtained thunderstorms lasting several hundreds of seconds, which is significantly higher than the duration of single thunderstorms. Therefore, our method identifies clusters of contiguous thunderstorms instead of single thunderstorms and prevent us comparing the evolution of our clusters with the evolution of singles thunderstorms defined by Emersic et al. (MWR, 2011). **Following the considerations of the reviewer and this reasoning, we have removed section 3.4.3.**

7.  In lines 469-479 the authors discuss the comparison between LIW maps and LCC(>20 ms) maps. There are several places where we see fires, but no strong lightning. What can be the interpretation of these fire events? Is there a possibility that those fires had been ignited by "regular" strokes, or those with shorter CC? It seems that the selection of 20ms threshold is arbitrary, and actually there may be episodes that even shorter strokes can ignite fires (for example if the forest was dry or already deteriorated).

    The lightning data employed to generate the map of LCC(>20 ms) flashes are reported by ISS-LIS during 4 years. Given that ISS-LIS orbits the Earth in a Low-Earth Orbit (LEO) and that the global frequency rate of occurrence of LCC(>20 ms) flashes is low (see Bitzer (JGR, 2017)), the lightning data reported by ISS-LIS is not large enough to produce a reliable climatology. As a consequence, we cannot assure that LIW flashes over areas where ISS-LIS did not report any LCC flash were not ignited by LCC flashes.

    We agree with the reviewer that the selection of 20 ms threshold could be arbitrary. However, it is not completely arbitrary. Fuquay et al., (JGR, 1972) reported the optical duration of 11 fire-igniting lightning flashes from ground. The shorter flash had a duration of

40 ms, which can give an approximation to the needed duration of a flash to ignite a fire. We used 20 ms instead of 40 ms because we assume that space-based observations can miss a portion of the optical signal as a consequence of cloud absorption of light. In fact, Bitzer (JGR, 2017) compared the optical signature of a LCC flash reported by TRMM-LIS with the electromagnetic signature reported by the Huntsville Alabama Marx Meter Array (HAMMA) sensor. HAMMA reported a continuing phase duration of 22 ms, while TRMM-LIS reported an optical duration of 7-9 ms. As Bitzer (JGR, 2017) concludes, "This example also demonstrates that the duration of continuing current determined from LIS measurements should be considered a minimum." **We have included this reasoning in the manuscript.**

Even with all the discussed uncertainties, we think this study is valuable. To the best of our knowledge, this is the first study that correlates LIW and LCC over the Mediterranean basin. We think the results of this study can be beneficial for the design of methods to scientifically exploit future missions, such as the MTG satellites (as we explain in our manuscript). The continuous monitoring of LCC flashes by MTG satellites will provide an unique opportunity to establish the threshold in the duration of flashes that can ignite fires in the studied region.

8. The discussion about forecasting the potential for LIW (lines 565-575) may benefit from including the concept of the Lightning Potential Index (LPI; Yair et al., JGR 2010). This parameter was later developed into the Dynamic Lightning Index by Lynn et al. (WAF, 2012). Perhaps simulating LIW events and "calibrating" the LPI values against the occurrence of LCC(>20 ms) will improve forecast capabilities in operational models.

We thank the reviewer for his/her suggestion to improve the discussion. **We have included it reasoning in the manuscript.**

---

## Author Comment (AC2)

**Rebuttal**

We would like to thank the reviewers for their thoughtful comments and efforts towards improving our manuscript. We address comments specific to reviewer 2 below (blue letters).

**Reviewer 2**

The paper elucidates the role of Long-Continuing-Current (LCC) lightning flashes. (20ms) in Lightning Induced Fires (LIW) by an exhaustive analysis of satellite and meteorological reanalysis products. The goal is to identify parameterizations for LIW that are informed by LCC, and some current treatments are shown to be incorrect. Furthermore, the LIE/LCC connection has not been probed in the Mediterranean region that makes this study valuable. The results are described and aggregated well and will be valuable to the community.

We thank the referee for these encouraging comments.

My major concern is that the paper focuses entirely on the LCC and meteorology (P, T, CAPE, $H_2O$) and does not evaluate the state of the vegetation (e.g. how dry?, vapor pressure deficit, soil moisture). Is this important for ignition and/or propagation, will this promote more dry-lightning events regionally? I would lie to see a short discussion explaining the role of these lower frequency drought periods. Specifically, are there any changes in the records of LIW/LCC relations during drought years in the long records analyzed here. Can this explain some of the difference reported between the Iberian peninsula and Greece.

As the reviewer points out, the state of vegetation is important for ignition, arrival and survival phases of LIW. A deep analysis of the state of vegetation is out of the scope of this work. However, we have now included an analysis of the runoff index for LIW in the Iberian Peninsula and Greece. The product "ERA5 hourly data on single levels from 1979 to present" includes the runoff index, defined as:

"Some water from rainfall, melting snow, or deep in the soil, stays stored in the soil. Otherwise, the water drains away, either over the surface (surface runoff), or under the ground (sub-surface runoff) and the sum of these two is called runoff…. The units of runoff are depth in meters of water. ..." (see more in [https://cds.climate.copernicus.eu/cdsapp#!/dataset/reanalysis-era5-single-levels?tab=overview](https://cds.climate.copernicus.eu/cdsapp#!/dataset/reanalysis-era5-single-levels?tab=overview)).

We think the runoff index can be a good proxy for the state of vegetation, as it represents the amount of water contained in the soil.

For the Iberian Peninsula, the median runoff index for typical CG flashes over coniferous/mixed forests is $9 \times 10^{-5}$ m, while it is $2 \times 10^{-5}$ m for LIW. For Greece, the median runoff index for typical CG flashes over coniferous/mixed forests is $2 \times 10^{-4}$ m, while for LIW it is $8 \times 10^{-5}$ m. Therefore, as the reviewer points out, the state of vegetation is also an important factor for the occurrence of LIW. In addition, this analysis shows that both typical CG lightning and LIW in the Iberian Peninsula took place over drier conditions than in Greece. **This analysis has now been included in the manuscript.**

We have collected the annual median value of the runoff index for typical CG flashes over coniferous/mixed forests and the ratio of total number of LIW to total number of CG flashes over coniferous/mixed forests :

**Iberian Peninsula**

| Year | Runoff typical CG (m) | Total number of LIW / Total number of CG |
| --- | --- | --- |
| 2009 | $7 \times 10^{-5}$ | $14 \times 10^{-3}$ |
| 2010 | $12 \times 10^{-5}$ | $7 \times 10^{-3}$ |
| 2011 | $13 \times 10^{-5}$ | $9 \times 10^{-3}$ |
| 2012 | $9 \times 10^{-5}$ | $9 \times 10^{-3}$ |
| 2013 | $17 \times 10^{-5}$ | $8 \times 10^{-3}$ |
| 2014 | $10 \times 10^{-5}$ | $2 \times 10^{-3}$ |
| 2015 | $6 \times 10^{-5}$ | $3 \times 10^{-3}$ |

**Greece**

| Year | Runoff typical CG (m) | Total number of LIW / Total number of CG |
| --- | --- | --- |
| 2017 | $11 \times 10^{-5}$ | $4.1 \times 10^{-4}$ |
| 2018 | $19 \times 10^{-5}$ | $2.8 \times 10^{-4}$ |
| 2019 | $17 \times 10^{-5}$ | $3.8 \times 10^{-4}$ |

Let us now analyze these data. The total number of LIW per lightning can be influenced by the detection efficiency of each lightning location system. Therefore, we plot these data after splitting them into 3 subgroups: WWLLN data over the Iberian Peninsula (2009-2013), ENTLN data over the Iberian Peninsula (2014-2015) and ENTLN data over Greece:

[Figure]

*Figure 1:*
*Annual ratio of total number of LIW to typical CG over coniferous and mixed forests in the Iberian Peninsula versus the median runoff index (m) for typical CG. These data correspond to the period between 2009 and 2013.*

[Figure]

*Figure 2:*
*Annual ratio of total number of LIW to typical CG over coniferous and mixed forests in the Iberian Peninsula versus the median runoff index (m) for typical CG. These data correspond to the period between 2014 and 2015.*

[Figure]

*Figure 3:*
*Annual ratio of total number of LIW to typical CG over coniferous and mixed forests in Greece versus the median runoff index (m) for typical CG. These data correspond to the period between 2017 and 2019.*

Figures 1-3 suggest a negative correlation between the annual total number of LIW per CG lightning and the annual median value of the runoff index in the Iberian Peninsula and Greece.

The ratio of the total number of LCC-lightning flashes to typical flashes reported by ISS-LIS in Europe for 2017, 2018 and 2019 (May-September) is, respectively, 0.01510884, 0.01211159 and 0.01143837. However, as we stayed in the manuscript, the total number of LCC-lightning flashes within the Iberian Peninsula and Greece is too low to analyze them at a regional scale.

Minor edits

58 LIW tend to occur in Clouds with High Base (CBH, prefer to high-base clouds, at multiple places in paper)

Done.

60 have been made

Done

70 RS are composed of a we identify shared meteorological conditions

Done